# Stabilizing non-iridium active sites by non-stoichiometric oxide for acidic water oxidation at high current density

Lingxi Zhou[1,4], Yangfan Shao[2,4], Fang Yin[2], Jia Li [2] ✉, Feiyu Kang[2,3] & Ruitao Lv [1,3] ✉

Stabilizing active sites of non-iridium-based oxygen evolution reaction (OER) electrocatalysts is crucial, but remains a big challenge for hydrogen production by acidic water splitting. Here, we report that non-stoichiometric Ti oxides ($TiO_x$) can safeguard the Ru sites through structural-confinement and charge-redistribution, thereby extending the catalyst lifetime in acid by 10 orders of magnitude longer compared to that of the stoichiometric one ($Ru/TiO_2$). By exploiting the redox interaction-engaged strategy, the in situ growth of $TiO_x$ on Ti foam and the loading of Ru nanoparticles are realized in one step. The as-synthesized binder-free $Ru/TiO_x$ catalyst exhibits low OER overpotentials of 174 and 265 mV at 10 and 500 mA cm$^{-2}$, respectively. Experimental characterizations and theoretical calculations confirm that $TiO_x$ stabilizes the Ru active center, enabling operation at 10 mA cm$^{-2}$ for over 37 days. This work opens an avenue of using non-stoichiometric compounds as stable and active materials for energy technologies.

Water electrolysis using renewable electricity has been regarded as a promising route to sustainable green hydrogen energy production[1–3]. Compared with traditional alkaline electrolysis, proton exchange membrane water electrolysis (PEMWE) has the advantages of faster dynamics, operation at higher current densities (maximum 2-3 A cm$^{-2}$), and high-purity $H_2$ production (>99.9999 vol%)[3–5]. However, the anodic oxygen evolution reaction (OER) is a four-proton-coupled-electron transfer process, which is the efficiency bottleneck of the water-splitting reaction due to a higher reaction energy barrier than the hydrogen evolution reaction (HER). In addition, the harsh acidic and strong oxidative operating conditions impose a huge challenge to the development of high-performance electrocatalysts for the acidic OER process, which hinders the widespread application of PEMWE[6,7]. More importantly, the use of abundant natural seawater as an electrolyte in water-splitting devices is highly desired but remains challenging due to the progressively increasing surface acidity at the anode due to the hydroxide removal during OER and hypochlorite formation[8,9].

Therefore, overcoming the activity/stability trade-off of acidic OER catalysts can also significantly address the challenges of natural seawater electrolysis.

For decades, Ir- and Ru-based materials have been considered as benchmark electrocatalysts to balance the stability and activity in acidic electrolyte[10,11]. However, the Ir-based materials suffer from low mass activity and high cost (US\$60,670 kg$^{-1}$), which has severely hindered its practical applications in a large scale; while Ru (US\$9,523 kg$^{-1}$) is 1/6 of the cost of Ir and typically exhibits higher intrinsic activity, making the Ru-based catalysts as a more ideal candidate to balance cost and activity in commercial acid-based devices[7,12–14]. Despite the above advantages, the use of Ru-based catalysts for acidic OER is hampered by the poor stability (<100 hours at 10 mA cm$^{-2}$) in acidic conditions due to the over-oxidation of Ru to high valence Ru$^{n+}$ (n > 4) materials (e.g. $RuO_4$) in highly oxidative environments, which inevitably leads to the collapse of the crystal structure[4,5,10,11], dissolution of active sites and thus deterioration of catalytic performance[4–7,10–14].

[1]State Key Laboratory of New Ceramics and Fine Processing, School of Materials Science and Engineering, Tsinghua University, Beijing 100084, China. [2]Institute of Materials Research and Shenzhen Geim Graphene Center, Tsinghua Shenzhen International Graduate School, Tsinghua University, Shenzhen 518055, China. [3]Key Laboratory of Advanced Materials (MOE), School of Materials Science and Engineering, Tsinghua University, Beijing 100084, China. [4]These authors contributed equally: Lingxi Zhou, Yangfan Shao. ✉e-mail: li.jia@sz.tsinghua.edu.cn; lvruitao@tsinghua.edu.cn

Therefore, to achieve high-performance OER over Ru-based catalysts, a trade-off between the activity and the stability is highly desirable and yet challenging. Recently, much effort has been devoted to improving the acidic OER performance of Ru-based electrocatalysts via strategies such as alloying[15,16], defect engineering[3,5,7,14,17], strain effect[4,18], structure tuning[13,14,19] and so on. These strategies aim to prevent excessive oxidation and dissolution of Ru active sites by tuning the electronic structure of Ru, resulting in optimized OER activity and stability compared to commercial $RuO_2$ nanoparticles in acids. Nevertheless, most of the reported catalysts are powder-based and need to be loaded onto substrates with the assistance of binders to prepare working electrodes[4–7,12–19]. The catalysts requiring binders suffer from high mass transport resistance, severe performance degradation, and easy delamination, especially at high current densities[20,21]. Therefore, most of the Ru-based catalysts reported to date can only operate at low current densities ($<100\,mA\,cm^{-2}$)[3–7,13–17,19], and the stability remains limited to within tens of hours at low current densities ($\sim 10\,mA\,cm^{-2}$)[4,5,12–14,17–19], which is far from meeting the industrial application requirements.

In this contribution, we report a one-step method for the in situ growth of $TiO_x$ nanorods on a Ti foam substrate while reducing $Ru^{3+}$ to small-sized Ru nanoparticles anchored thereon ($Ru/TiO_x$) as a robust and efficient binder-free electrode for OER in acidic media. By applying this cost-effective method, which is free of additional Ti source, reducing agent, and binder, we were able to construct intrinsic defects in the $TiO_x$ (x denotes the non-stoichiometric compound resulting from oxygen vacancies), which stabilized the highly active Ru sites through the enhanced metal-support interaction. As a result, the $Ru/TiO_x$ exhibits low overpotentials of 174, 209, and 265 mV to achieve current densities of 10, 100, and even a high current density of $500\,mA\,cm^{-2}$ in a challenging acid electrolyte, respectively. The non-stoichiometric oxide ensures the charge accumulation at the Ru site, resulting in a significant improvement of OER activity (2.9 times) and stability (10.0 times), compared to the stoichiometric $RuO_x/TiO_2$. More importantly, $Ru/TiO_x$ can be directly applied to the electrolysis of pure natural seawater, requiring an overpotential of 320 mV to achieve a high current density of $100\,mA\,cm^{-2}$, suggesting its great potential for practical seawater-splitting applications. Therefore, we offer an integrated preparation strategy to overcome the activity/stability limitation of Ru-based catalysts. The non-stoichiometric compound-induced charge redistribution at metal sites demonstrated in this work may pave wide avenues for the development of promising catalysts for practical PEMWEs and other acid-based devices.

## Results

### Synthesis and characterization of catalysts

The synthesis process of the self-supported $Ru/TiO_x$ electrode is schematically shown in Fig. 1a. Through a facile hydrothermal process, the in situ growth of $TiO_x$ and the loading of Ru nanoparticles (NPs) are realized in one step. Specifically, commercial Ti foam (TF) with a porous structure and good chemical/mechanical stability is selected as both substrate and Ti source for $TiO_x$ growth. The Ti foam serves as both substrate and reducing agent to efficiently reduce $Ru^{3+}$ ions through a redox interaction-engaged strategy due to a thermodynamically favorable process. Photographs (Supplementary Fig. 1) show that $Ru/TiO_x$ is completely covered on TF, and the color of TF changes from gray to dark. Typically, aqueous solutions of $RuCl_3$ and HCl were heated up to 200 °C for 20 h. HCl not only acted as a chemical corrosive agent to induce defects on the TF surface, which enabled the growth of rutile phase nanorods, but also provided an acidic condition to etch the TF and form titanium ions ($Ti^{3+}$), which were transformed into titanium (III) (hydro)oxide nanostructure via $Ti^{3+}$ hydrolysis in acidic aqueous solution during the hydrothermal treatment[22]. Subsequently, when the metal ions ($Ru^{3+}$) came into contact with the titanium(III) (hydro)oxide nanostructure, which had a strong reducing ability, they were immediately reduced by the $Ti^{3+}$ and

rapidly nucleated on the as-formed $TiO_2$ nanorod surfaces to grow into clusters and further into nanoparticles. Meanwhile, the support was partially oxidized by the metal ions and further converted into $TiO_x$, resulting in $Ru/TiO_x$ nanocomposites[23,24].

Field emission scanning electron microscopy (FESEM; Supplementary Figs. 2, 3) reveals the 3D porous structure of pristine TF and the successful growth of $TiO_x$ nanorod arrays. Transmission electron microscopy (TEM) and corresponding energy dispersive X-ray spectroscopy (EDS) mapping profiles reveal the high dispersion of Ru NPs, and homogeneous distribution of Ti and O elements throughout the $Ru/TiO_x$ sample (Fig. 1b). According to high-resolution TEM (HRTEM; Supplementary Fig. 4), Ru NPs with ~ 4 nm are homogeneously deposited on $TiO_x$, which has a rod-like structure with pyramidal morphology consisting of dominantly exposed reductive lateral facets (110) and oxidative top facets (111). The anisotropic growth of rutile nanocrystals along the [001] direction was caused by excess $Cl^-$ preferentially adsorbed on the (110) plane of the rutile $TiO_2$ surface, which suppressed grain growth along the [110] direction and accelerated it in the [001] direction[22,25]. The d-spacing of the nanorods is determined to be 0.32 nm, corresponding to the (110) crystal plane fringes of the rutile $TiO_2$; the d-spacing of the nanoparticles is determined to be 0.21 nm, attributed to the Ru (101) plane, which agrees well with the negligible broad peaks of Ru due to the high Ru dispersion in the X-ray diffraction (XRD) pattern (Supplementary Fig. 5). The high-angle annular dark-field scanning TEM (HAADF-STEM) image and the intensity maps clearly show the atomic arrangement of Ru, in which the lattice space of 0.23 nm corresponds to the Ru {100} family of crystal planes (Fig. 1e and Supplementary Fig. 6). Combined with STEM-EDS (Fig. 1b, c), it can be explained that Ru NPs with the size of ~4 nm are homogeneously deposited on in situ formed rutile $TiO_x$ nanorod by one-step hydrothermal process. Interestingly, it is found that the atomic arrangement at the edge of $TiO_x$ nanorod is different from that of the body part. As shown in Fig. 1d, rutile-type $TiO_2$ (above the interface) is in the body part of the nanorod, while $Ti_2O_3$ (lies below) is at the edge, assuming a zigzag configuration, which is typical of corundum-type structure[26–28]. According to the previously proposed growth mechanism, the redox reaction between titanium(III) (hydro)oxide and Ru ions during the hydrothermal process may be responsible for the formation of the $Ti_2O_3$ phase at the edge[24]. For comparison, we synthesized $TiO_2$ on TF substrate using the same method (except for the addition of Ru source). The FESEM images show the nanorod structure of $TiO_2/TF$ with flat apex, which is different from that of the $Ru/TiO_x$ with pyramidal apex (Supplementary Fig. 7). The HAADF-STEM image and the corresponding fast Fourier transforms (FFTs) show the rutile-type $TiO_2$ with (101) crystal plane in both the body and the edge part of the nanorod (Supplementary Fig. 8). Therefore, the redox reaction during the hydrothermal process results in the formation of a highly active edge part of the nanorod, which leads to the coexistence of $TiO_2$ and $Ti_2O_3$ in the as-synthesized sample, which we denoted as $Ru/TiO_x$. It is reported that the sub-stoichiometric phase $Ti_2O_3$ exhibits high electrical conductivity and the rutile-corundum interface is favorable for charge transfer, which is beneficial in energy storage systems and electronic devices[26–28]. Therefore, we believe that the in situ formed $Ru/TiO_x$ with supported acid-stable Ti framework, ultrafine nanoparticles, and rutile-corundum interface can be a good candidate for OER electrocatalysis in acidic media.

### Electrocatalytic OER performance in acidic media

The catalytic performances of self-supported $Ru/TiO_x$ towards OER were directly evaluated in $0.5\,M\,H_2SO_4$. For comparison, annealed $RuO_x/TiO_2$, $TiO_2/TF$, commercial $RuO_2/TiO_2$, and $IrO_2/TiO_2$ electrocatalysts were also investigated under the same conditions. We compare the OER activity in three prevalent metrics, i.e., geometric activity ($j_{geo}$, geometric area of the working electrode), specific activity

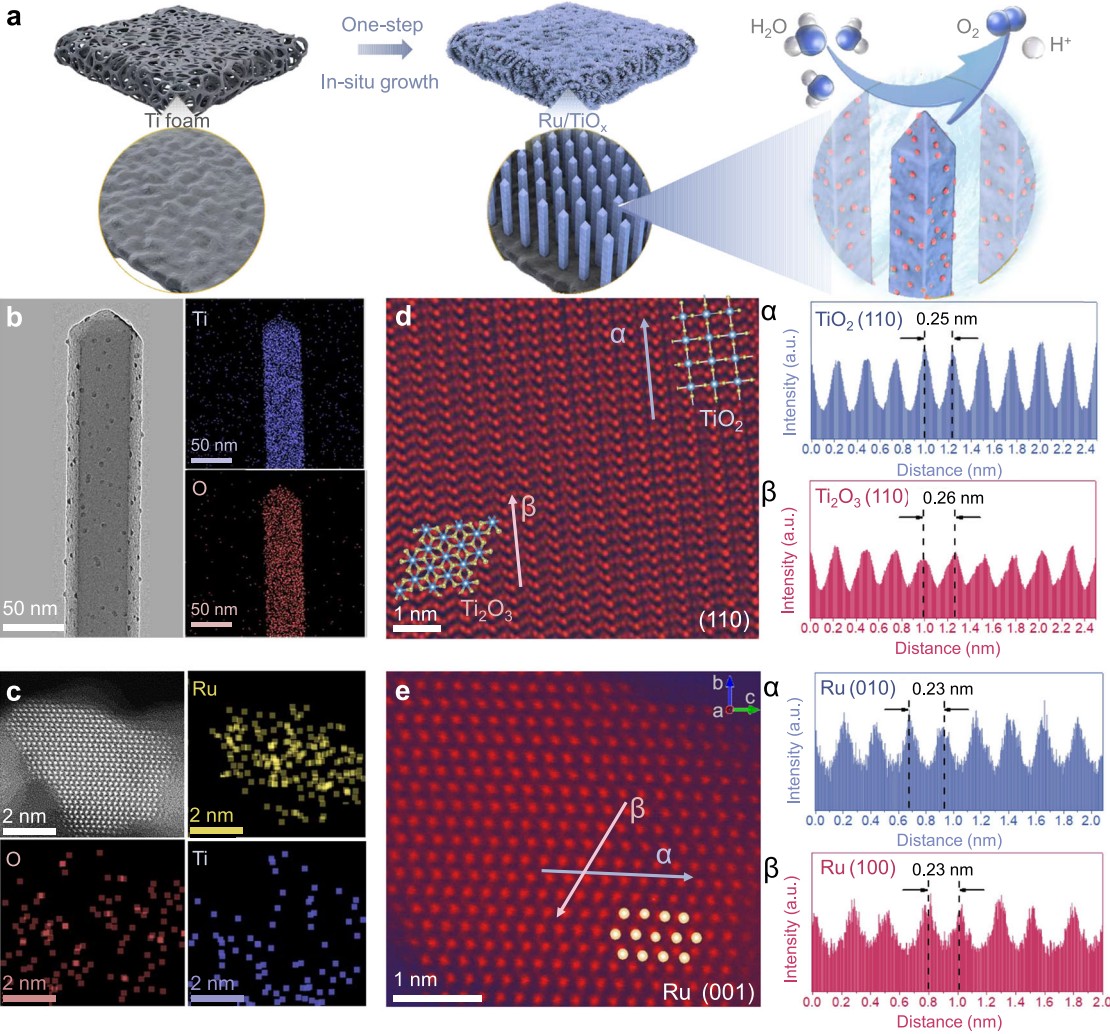

**Fig. 1 | Morphology and structure characterizations of as-synthesized Ru/TiO$_x$.**
**a** Schematic illustration of synthesis of Ru/TiO$_x$ as binder-free electrode towards oxygen evolution reaction (OER) in acidic media. **b** Transmission electron microscopy (TEM) image of Ru/TiO$_x$ and corresponding energy dispersive X-ray spectroscopy (EDS) mapping profile of Ti (blue) and O (red). **c** Aberration-corrected high angle annular dark field-scanning TEM (HAADF-STEM) image of Ru/TiO$_x$ and corresponding EDS mapping profile of Ru (yellow), Ti (blue) and O (red).
**d, e** Aberration-corrected HAADF-STEM images of TiO$_x$ support (**d**) and Ru site of Ru/TiO$_x$ (**e**). STEM intensity profiles are presented to directions labeled with blue (α) and red (β) arrows. (α and β) show the corresponding intensity profiles and the lattice fringe spacing.

($j_s$, electrochemical active surface area (ECSA)), and mass activity ($j_m$, mass of metals). Geometric activity is important at the device level. As can be seen from the *iR*-free linear sweep voltammetry (LSV) curves, Fig. 2a, the Ru/TiO$_x$ exhibits the highest performance of all samples, requiring overpotentials of only 174, 209, and 226 mV to achieve current densities of 10, 100, and even 200 mA cm$^{-2}_{geo}$, respectively, significantly outperforming the other control samples, including the state-of-the-art commercial RuO$_2$/TiO$_2$ and IrO$_2$/TiO$_2$. To eliminate electrolyte resistance, the LSV curves with 95 % *iR*-compensation were also shown in Fig. 2a. The measured overpotentials before and after *iR* correction are summarized in Supplementary Fig. 9. Supplementary Movie 1 shows the process of the binder-free Ru/TiO$_x$ electrode working directly as an anode for water electrolysis at different current densities. It can be observed that vigorous oxygen bubbles are rapidly released from the anode. In addition, under the harsh acidic and oxidative environment, no catalyst is detached from the electrode during the whole process of the stability test, proving its superiority in high current density electrolysis. The trend of overpotentials to achieve a high current density of 100 mA cm$^{-2}$ is ranked in the following order: Ru/TiO$_x$ (209 mV) <annealed RuO$_x$/TiO$_2$ (232 mV) <commercial RuO$_2$/TiO$_2$ (269 mV) <commercial IrO$_2$/TiO$_2$ (345 mV)

(Fig. 2d). The bare in situ formed TiO$_2$ nanorods grown on TF (TiO$_2$/TF) under the same hydrothermal conditions showed negligible OER performance, while commercial RuO$_2$/TiO$_2$ ($\eta_{10}$ = 204 mV) showed improved electrocatalytic activity than commercial RuO$_2$/TF ($\eta_{10}$ = 242 mV) at the same loading, suggesting that TiO$_2$ itself is inert in OER but plays an important role in supporting and dispersing active sites (Supplementary Fig. 10). Compared with the TiO$_2$-based samples, the physically-adsorbed RuO$_2$/TiO$_2$ shows significantly lower OER activity than both the in situ formed Ru/TiO$_x$ and the annealed RuO$_x$/TiO$_2$ electrode, demonstrating the superiority of the in situ growth strategy, which results in a strong internal interaction between Ru and TiO$_x$, thereby optimizing the intrinsic activity of each active site. The OER performance of the Ru/TiO$_x$ is firstly compared in terms of overpotential to reach 10 mA cm$^{-2}$. The OER activity of Ru/TiO$_x$ is substantially superior to that of most Ru/Ir-based electrocatalysts reported in the literature (Fig. 2e and Supplementary Table 1), including recently reported representative electrocatalysts Ni-RuO$_2$ (214 mV)[3], Ru/Co-N-C (232 mV)[29], etched-Ru/Fe oxide nano assemblies (238 mV)[30], Ru@IrO$_x$ (282 mV)[31], RuNi$_2$@graphene (227 mV)[12], and so on. More importantly, the excellence of electrocatalytic performance of Ru/TiO$_x$ is more pronounced at high current densities. For example,

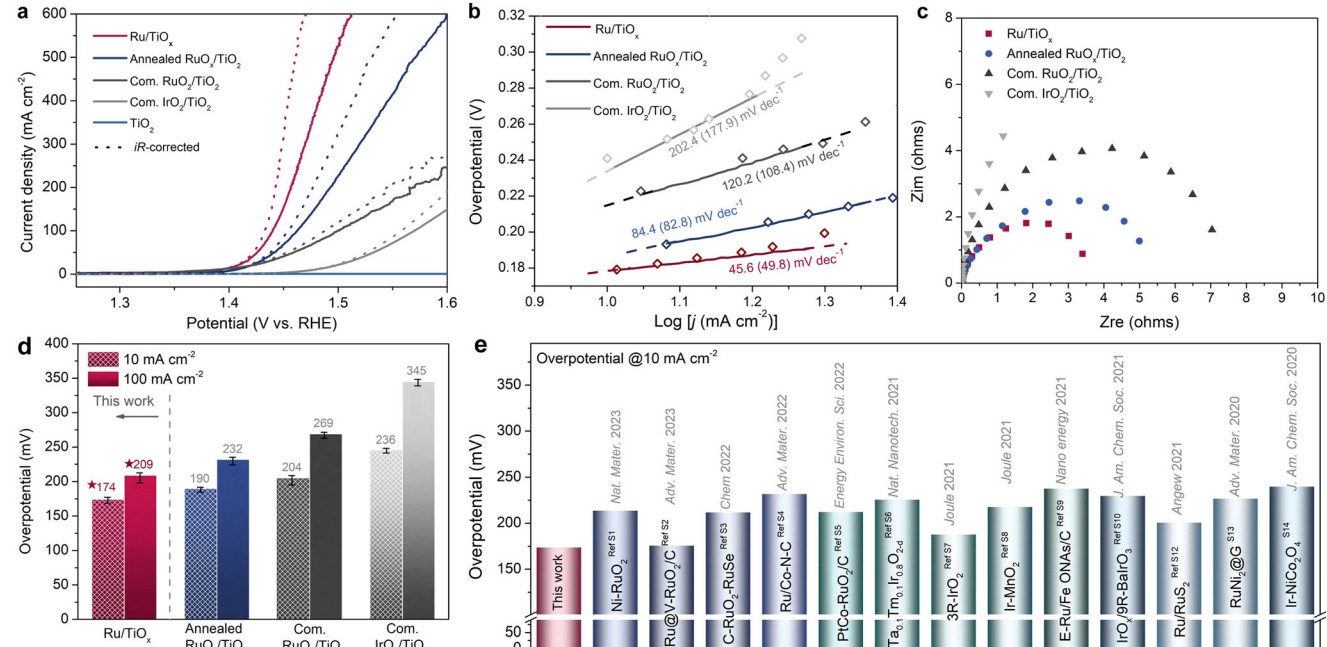

**Fig. 2 | Electrocatalytic oxygen evolution reaction (OER) activity in 0.5 M H₂SO₄ solutions (pH = 0.3). a** Linear sweep voltammetry (LSV) curves (both *iR*-corrected and *iR*-free) of Ru/TiOₓ, annealed RuOₓ/TiO₂, com. RuO₂/TiO₂, com. IrO₂/TiO₂ and TiO₂. (Com. RuO₂/TiO₂ and com. IrO₂/TiO₂ denotes commercial RuO₂/TiO₂ and commercial IrO₂/TiO₂, respectively). *iR* is automatically compensated by workstation (95%). Mass loadings of noble metals are 0.0715, 0.0867, 0.0992 and 0.1135 mg cm⁻² for Ru/TiOₓ, annealed RuOₓ/TiO₂, com. RuO₂/TiO₂ and com. IrO₂/TiO₂, respectively. **b** Tafel plots derived from the LSV curves (solid line) and the steady-state polarization curves (scatters) (values in parentheses were derived from steady-state polarization curves). **c** Electrochemical impedance spectroscopy (EIS) of Ru/TiOₓ, annealed RuOₓ/TiO₂, com. RuO₂/TiO₂ and com. IrO₂/TiO₂. **d** Comparison of overpotentials without *iR* correction at 10 and 100 mA cm⁻² for Ru/TiOₓ, annealed RuOₓ/TiO₂, com. RuO₂/TiO₂ and com. IrO₂/TiO₂. (Error bar: standard error of three repeated measurements). **e** Comparison of the overpotentials of Ru/TiOₓ and state-of-the-art Ru/Ir-based electrocatalysts at 10 mA cm⁻² in acidic media.

the Ru/TiOₓ electrode requires an overpotential of only 265 mV to achieve a high current density of 500 mA cm⁻², which is required by the industrial criteria but rarely achieved in the acidic OER process[3–7,13–17,19]. It is well known that the specific activity is mainly determined by the ECSA of the catalysts. In order to evaluate the corresponding intrinsic activity, the electrochemical double-layer capacitance ($C_{dl}$) was used to compare the order of the ECSA. Clearly, Ru/TiOₓ exhibits a much higher $C_{dl}$ value of 13.73 mF/cm2, revealing the exposure of more active sites for OER (Supplementary Fig. 11). We further normalized the LSV curves with ECSA to compare the specific activity of the electrocatalysts (Supplementary Fig. 12a). Interestingly, the specific activities and the geometric activity show the same trend (Supplementary Fig. 12b). The Ru/TiOₓ can reach a high $j_{ECSA}$ of 1.49 mA cm⁻² at 1.50 V vs. RHE, which is about 2.5 times that of the Com-RuO₂/TiO₂ (about 0.60 mA cm⁻²), indicating its superior intrinsic activity for the OER. To quantitatively evaluate the mass activity, we determined the corresponding mass loading of the Ru/TiOₓ and other control samples by inductively coupled plasma mass spectrometry (ICP-MS) measurement (Supplementary Table 2). As shown in Supplementary Fig. 13, the mass activity of Ru/TiOₓ (2128.2 A g$_{Ru}$⁻¹) at the potential of 1.45 V vs. RHE is 4.6 and 46.2 times that of Com-RuO₂/TiO₂ (462.0 A g$_{Ru}$⁻¹) and Com-IrO₂/TiO₂ (46.0 A g$_{Ir}$⁻¹), respectively. These results further substantiated the remarkable electrocatalytic performance of Ru/TiOₓ compared to the other reported catalysts (Supplementary Table 3). The intrinsic activities of Ru/TiOₓ were further assessed based on turnover frequencies (TOFs) at different overpotentials, which are among the highest when compared with representative OER catalysts in various acidic media (Supplementary Fig. 14a and Table 4). For example, the TOF of Ru/TiOₓ is calculated to be 1.960 s⁻¹ at an overpotential of 300 mV based on the total loading mass, which increases to 2.192 s⁻¹ when calculated based on ECSA (Supplementary Table 5). We also provided a bar graph of ECSA-normalized current densities

and TOF values at an overpotential of 300 mV, which shows a similar activity trend (Supplementary Fig. 14b). The above results prove that Ru/TiOₓ shows the highest intrinsic activity per site. In order to evaluate the catalytic kinetics of OER, Tafel plots are obtained based on the *iR*-free LSV curves and the steady-state polarization curves (Fig. 2b and Supplementary Fig. 15)[14], where Ru/TiOₓ exhibits the lowest Tafel slope of 45.6 (49.8) mV dec⁻¹, indicating the highest charge transfer efficiency and fastest reaction rate among these prepared samples. The electrochemical impedance spectroscopy (EIS) measurements also confirm a faster OER process of Ru/TiOₓ, as evidenced by its remarkably lower charge transfer resistance ($R_{ct}$) than that of other control samples (Fig. 2c). From the above electrochemical results, it is clear that the well-aligned Ru/TiOₓ nanoarrays play an important role in the catalyst-liquid/gas interaction at the interfaces and consequently in the OER performance, especially at high current densities. To quantitatively analyze the differences between the TF and the Ru/TiOₓ, we measured the liquid contact angles (LCA) and bubble contact angles (BCA) on their surfaces, respectively. The well water adsorption originating from the Ru/TiOₓ nanoarrays significantly improves the surface wettability and then favors the contact between the electrolytes and the electrode surface, as evidenced by the decreased LCA from 43.6° (TF) to 0° (Ru/TiOₓ) (Supplementary Fig. 16). In addition, the vertically aligned nanoarrays result in effective gas escape, especially at high current densities, due to the discontinuous state of the three-phase gas-solid-liquid contact line between the bubbles and the electrode surfaces, which contributes to the exceptionally small contact area and lower adhesion force[32]. Therefore, the Ru/TiOₓ shows a higher BCA of 151.3°compared to the TF (Supplementary Fig. 17), which shows a more tremendous potential to release the as-formed O₂ bubbles with superfast speed and prevent bubble retention.

Long-term stability under high current density is another essential standard for evaluating industrial applications, which is particularly

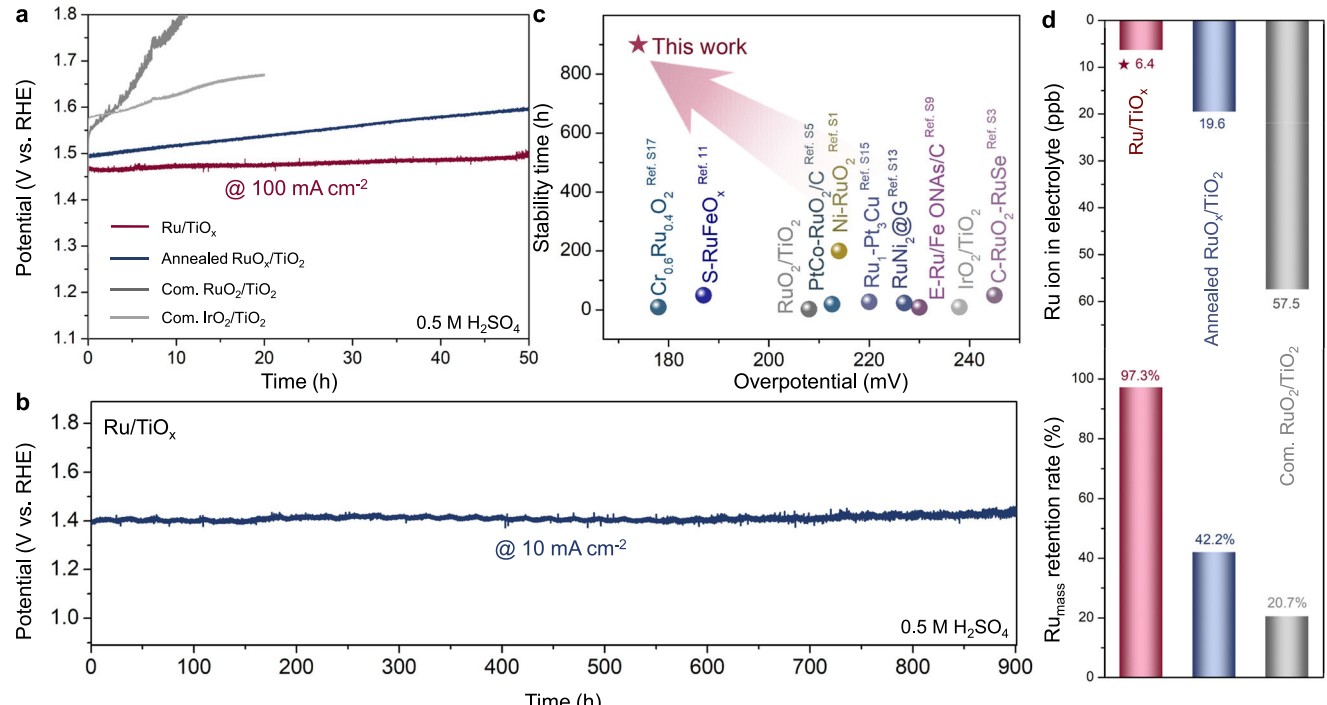

**Fig. 3 | Electrocatalytic OER stability in 0.5 M $H_2SO_4$ solutions (pH = 0.3).**
**a** Chronoamperometric curves of Ru/TiO$_x$, annealed RuO$_x$/TiO$_2$, com. RuO$_2$/TiO$_2$ and com. IrO$_2$/TiO$_2$ for OER at 100 mA cm$^{-2}$. The chronoamperometric curves are not *iR* compensated. Mass loadings of noble metals are 0.0715, 0.0867, 0.0992, and 0.1135 mg cm$^{-2}$ for Ru/TiO$_x$, annealed RuO$_x$/TiO$_2$, com. RuO$_2$/TiO$_2$ and com. IrO$_2$/TiO$_2$, respectively. **b** Chronoamperometric curves of Ru/TiO$_x$ for OER at 10 mA cm$^{-2}$.

**c** Comparison of the overpotential and stability time of Ru/TiO$_x$ with state-of-the-art OER electrocatalysts in acidic media. **d** Inductively coupled plasma-mass spectrometry (ICP-MS) analysis for dissolved Ru ions in post chronopotentiometry electrolyte and Ru mass percentage retained in Ru/TiO$_x$, annealed RuO$_x$/TiO$_2$ and com. RuO$_2$/TiO$_2$ catalyst after the chronoamperometric test.

critical for acidic OER electrocatalysts due to the highly corrosive electrolytes and oxidative operating conditions. Thus, the OER stability of the as-prepared Ru/TiO$_x$ was evaluated by continuous cyclic voltammograms (CVs) up to 550 mA cm$^{-2}$ for 50 cycles and chronopotentiometry test at constant current densities of 10 mA cm$^{-2}$ and 100 mA cm$^{-2}$. As shown in Supplementary Fig. 18, no obvious decay in polarization curves was observed for Ru/TiO$_x$ after 50 OER cycles up to 550 mA cm$^{-2}$, suggesting its durability under large current densities. Furthermore, the overpotential required to achieve the current density of 100 mA cm$^{-2}$ remained constant with a negligible increase of 28 mV over 50 h (Fig. 3a). In contrast, the annealed RuO$_x$/TiO$_2$ electrode exhibited a continuously increasing overpotential of 92 mV until the final activity degradation, while the commercial RuO$_2$/TiO$_2$ and IrO$_2$/TiO$_2$ underwent severe OER performance decay within only 20 h. Similar to the control samples, many other reported catalysts suffer from poor stability of only a few hours at low current density (10 mA cm$^{-2}$) due to the loss of active phase under this harsh condition (Supplementary Fig. 19 and Table 1). However, the as-synthesized Ru/TiO$_x$ showed remarkable long-term stability for OER at the current density of 10 mA cm$^{-2}$. A nearly horizontal line with only a 20 mV increase in overpotential was obtained after 900 h of continuous CP test, maintaining 98.6% of the initial activity, further confirming the high stability of our catalysts in acidic environment (Fig. 3b). Notably, the low overpotential of 174 mV and the long-term durability of 900 h at 10 mA cm$^{-2}$ for the Ru/TiO$_x$ surpasses most of the reported acidic OER electrocatalysts (Fig. 3c and Supplementary Table 1).

ICP-MS analysis and X-ray photoelectron spectroscopy (XPS) were further performed to determine the amounts of dissolved Ru ions in the electrolytes after the stability test and the Ru content remained in the catalysts (Fig. 3d). For the as-synthesized Ru/TiO$_x$, the extremely low Ru ion concentration (6.4 ppb) in the electrolyte after the stability test and the maintenance of the Ru content (97.3%) in the catalyst manifest

the effective protection of Ru sites from dissolution (Fig. 3d and Supplementary Table 2). For comparison, only 42.2% and 20.7% Ru remained in the annealed RuO$_x$/TiO$_2$ and commercial RuO$_2$/TiO$_2$ catalyst, proving the stability of the Ru sites in the Ru/TiO$_x$ catalyst, consistent with the trend obtained by XPS results (Supplementary Fig. 20 and Table 2).

To illustrate the practical capabilities of Ru/TiO$_x$ in water splitting, a PEMWE was constructed to evaluate the performance under conditions that are representative of industrial applications (Supplementary Figs. 21 and 22). As shown in Supplementary Fig. 22a, the PEMWE with Ru/TiO$_x$ as an anode shows 1.71 V at 1 A cm$^{-2}$, which is 0.23 V lower compared with RuO$_2$ anode. Besides, the PEMWE with Ru/TiO$_x$ as an anode well maintains its voltage ($\Delta E < 0.01$ V) during 200 h operation at 500 mA cm$^{-2}$, comparable to the recently reported catalysts (Supplementary Fig. 22b and Table 6). In addition, the stability significantly outperforms those with RuO$_2$ anode, which shows $\Delta E > 0.2$ V decay within 100 h (Supplementary Fig. 22b), further highlighting the superiority of self-supported Ru/TiO$_x$ in OER at high current densities.

## Electrocatalytic OER performance in natural seawater
In particular, in the seawater electrochemical splitting process, the pH at the anode decreases dramatically, resulting in a local acidic environment and deactivation of most electocatalysts[8,33]. Therefore, an additional buffer solution such as KOH is required to mitigate the corrosion and dissolution of the catalyst caused by this local acidity in the seawater electrolysis[34,35]. However, the use of such chemical agents increases the excess consumption and cost, especially in large-scale applications. From a green and sustainable point of view, it is highly desirable to directly use abundant natural seawater without any treatment in water electrolyzer as an energy-efficient technology, but it still remains a great challenge. Therefore, the search for an active and durable anode material that can survive in natural seawater, rather than buffered or simulated seawater, is of great importance[33].

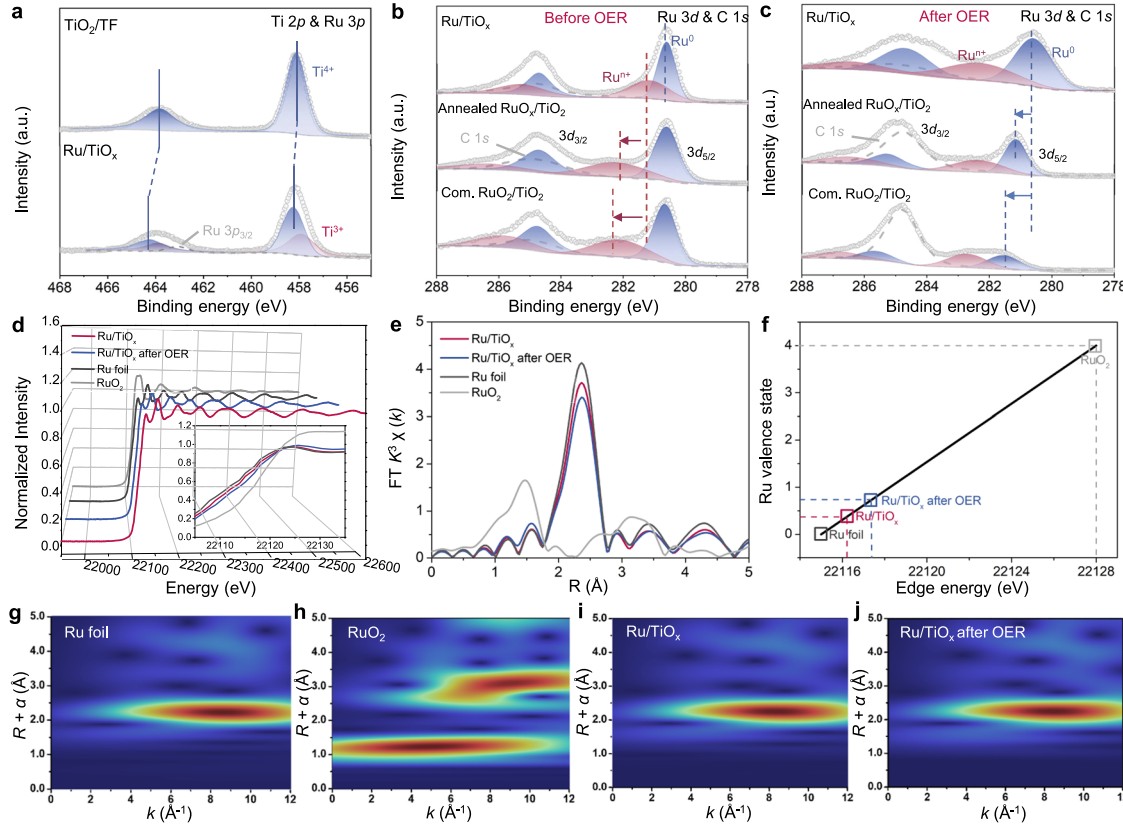

**Fig. 4 | Electronic structure analysis of Ru/TiOₓ. a** XPS of Ti 2*p* and Ru 3*p* for the bare TiO₂ and Ru/TiOₓ. **b, c** Ru 3*d* XPS spectra of Ru/TiOₓ, annealed RuOₓ/TiO₂ and com. RuO₂/TiO₂ before OER stability test (**b**) and after OER stability test (**c**). **d** Ru *K*-edge synchrotron-based X-ray absorption near-edge structure (XANES) spectra of Ru/TiOₓ before and after OER stability test using Ru foil and commercial RuO₂ as references. **e** Fourier-transformed (FT) *k*3-weighted *χ(k)*-function of the extended X-ray absorption fine structure (EXAFS) spectra for the Ru *K*-edge. **f** Relation between the Ru *K*-edge absorption energy ($E_O$) and valence states for Ru/TiOₓ, Ru/TiOₓ after OER stability test, Ru foil, and RuO₂. **g–j** Wavelet transforms for the *k*3-weighted EXAFS signals of Ru foil (**g**), RuO₂ (**h**), Ru/TiOₓ (**i**) and Ru/TiOₓ after OER stability test (**j**).

As a binder-free electrode, the as-synthesized Ru/TiOₓ exhibits good activity and durability under acidic and oxidizing conditions as concluded above. Based on the reduced-pH at the anode during the electrolysis of seawater, we believe that such a stable structure can also prevent the loss of catalytic sites in a similar local-acidic environment during OER in real seawater without any buffer solutions. As expected, the Ru/TiOₓ exhibited similarly high OER performance in real seawater (Yellow Sea, China) without any further treatment, requiring an overpotential of 320 mV to achieve a high current density of 100 mA cm⁻² (Supplementary Fig. 23a). More importantly, the Ru/TiOₓ maintained the initial high catalytic activity in seawater even after prolonged use. As shown in Supplementary Fig. 23b and Movie 2, Ru/TiOₓ can deliver a 100 mA cm⁻² at a constant potential of 1.57 V vs. RHE. Although there are minor fluctuations during the reaction process due to the complex composition of the unpurified and unbuffered seawater electrolyte (Supplementary Table 7), the performance remained almost unchanged at the end of the 20 h reaction. It is noteworthy that almost all HER and OER catalysts reported in seawater electrolysis require an additional buffer solution[34,35]. Even the very few reported catalysts that can be used in buffer-free seawater cannot achieve a high current density and remain stable for such a long time[9,36]. Therefore, the as-prepared catalyst represents a breakthrough in the direct utilization of natural water resources, which has great potential for large-scale applications of seawater-based energy storage and conversion devices.

## Origin of improved electrocatalytic OER performance
To elucidate the origin of the enhanced OER catalytic performance of the Ru/TiOₓ in an acidic environment, we synthesized two control

samples of annealed RuOₓ/TiO₂ and physically mixed RuO₂/TiO₂ with the same loading. The above results indicate that the two control samples exhibited apparently low electrocatalytic activity and stability in an acidic solution compared with those of Ru/TiOₓ. This motivates us to investigate the differences in the microstructure and electronic structure of Ru/TiOₓ, RuOₓ/TiO₂, and RuO₂/TiO₂, which are closely related to the electrocatalytic performance for acidic OER. The chemical states for Ru, Ti, and O in the three samples before and after the OER stability test were first investigated by XPS analysis.

In particular, the in situ formation of Ru/TiOₓ via one-step hydrothermal process induces a strong interaction between the Ru sites and the TiOₓ support, as evidenced by the positive core-level shift in the binding energies of Ti 2*p* and O 1*s* compared to bare TiO₂/TF (Fig. 4a and Supplementary Fig. 24). This interaction allows the generation of Ti³⁺ defects and oxygen vacancies, which are not only beneficial for preventing the oxidation of metallic Ru to soluble oxidized species (Ru^{>4+}), but also favorable for bonding with water molecules, promoting the water splitting process[9,37,38]. It should be noted that the annealing of Ru/TiOₓ resulted in the formation of RuOₓ with a mostly oxidized surface and the complete transformation of Ti₂O₃ to TiO₂, as indicated by the positive core-level shift of the binding energies of Ru 3*d* and Ti 2*p* (Fig. 4b and Supplementary Fig. 25). These changes suggest that the removal of the Ti³⁺ defects and oxygen vacancies weakened the interaction between Ru and TiOₓ, which led to the low electrocatalytic activity and stability of the annealed RuOₓ/TiO₂[37].

To further verify the high stability of Ru/TiOₓ for OER in the acidic electrolyte, the chemical states for Ru, Ti, and O in the three samples after the acidic OER stability tests were analyzed and compared with

those before OER (Fig. 4c, Supplementary Table 8). For Ru/TiO$_x$, the peak at 280.61 eV for Ru 3$d_{5/2}$ (Ru$^0$) remained generally unchanged as compared with that of the sample before the OER test (280.60 eV), while the peak at 281.2 eV for Ru 3$d_{3/2}$ (Ru$^{n+}$, n < 4) slightly shifted to higher binding energies, indicating that the active Ru sites were only partially oxidized but mainly remained in the low-valence state (Ru$^{n+}$, n < 4) during the 50 h OER test (Fig. 4c). For comparison, the changes for annealed RuO$_x$/TiO$_2$ and commercial RuO$_2$/TiO$_2$ are much more significant: the peaks for Ru$^0$ and Ru$^{4+}$ shift positively for 0.54 and 0.90 eV, respectively, indicating that the active species Ru were all over-oxidized to Ru$^{n+}$ (n > 4), which were easily separated from the catalysts and dissolved during the reaction process, leading to the degradation of the OER performance. The Ru$^{n+}$/Ru$^0$ ratios calculated from the corresponding peak area in the Ru 3$d$ XPS spectra indicate that the oxidation state of Ru in the as-prepared catalysts follows the trend of commercial RuO$_2$/TiO$_2$ > annealed RuO$_x$/TiO$_2$ > Ru/TiO$_x$, which is opposite to the trend of OER stability (Supplementary Fig. 26). The established valence-stability relationship proved that the low-valence Ru in Ru/TiO$_x$, due to the strong interaction between Ru and TiO$_x$, is highly active and stable, further highlighting the advantages of the in situ and one-step growth strategy. Afterwards, the chemical environments of Ti and Ru in the Ru/TiO$_x$ catalyst under actual OER conditions were monitored by in-situ XPS (Supplementary Figs. 27–29 and Table 9). Significantly, the Ru 3$d$ XPS peaks at 280.1 and 284.2 eV exhibit negligible changes with the applied potential increased from 1.0 to 1.7 V vs. RHE (Supplementary Fig. 28a and Table 9). Detailed quantitative analysis shows the ratio of Ru$^{n+}$/Ru$^0$ remains almost identical at 0.7 as the voltage increases, which is consistent with the ex-situ XPS analysis results (Supplementary Figs. 26 and 29). The above results further confirm the stable Ru chemical state in Ru/TiO$_x$ during the OER process. For the Ti 2$p$ spectra (Supplementary Figs. 28b and Table 9), the coexistence of Ti$^{3+}$ and Ti$^{4+}$ species was distinguished. Interestingly, as the voltage increases, the ratio of Ti$^{3+}$/Ti$^{4+}$ slightly decreases and remains stable at 0.17, indicating the critical role of the Ti$^{3+}$ in stabilizing Ru active sites. Corresponding to this phenomenon is the O$_V$/O$_L$ (oxygen vacancy/lattice oxygen) ratio obtained from the O 1$s$ spectra (Supplementary Figs. 28c and 29). It can be seen that as the voltage increases, the O$_V$/O$_L$ ratio first decreases and then almost returns to the initial state, manifesting that O$_V$ regeneration is accompanied by the release of oxygen, thereby stabilizing active species[39].

X-ray absorption spectroscopy (XAS) is also used to precisely investigate the electronic structure of Ru in the Ru/TiO$_x$. In detail, the X-ray absorption near edge structure (XANES) spectra (Fig. 4d) show that the adsorption threshold position is close to that of Ru foil but remarkably different from that of RuO$_2$, indicating that metallic Ru NPs are dominant in Ru/TiO$_x$. In addition, the Ru $K$-edge spectra before and after the OER reaction are quite similar (Fig. 4b–d), indicating that the oxidation state of Ru is stable, as suggested by the XPS results. Specifically, the Ru valence states are quantitatively measured by the adsorption energy ($E_O$) of the Ru/TiO$_x$ before and after OER (Fig. 4f and Supplementary Table 10). After the OER test, the $E_O$ of Ru/TiO$_x$ was positively shifted to higher energy compared to that of pristine Ru/TiO$_x$, with the oxidation state of Ru slightly increased from +0.34 to +0.72 during electrocatalysis, which is still far below the +4 valence state. Therefore, the strong interaction between Ru and TiO$_x$ effectively suppresses the over-oxidation of Ru during the OER, resulting in high catalytic stability. Subsequently, the corresponding Fourier transforms of the extended X-ray absorption fine structure (EXAFS) spectra show that the length of Ru-Ru bonds in Ru/TiO$_x$ is similar to that of Ru foil, which shows the dominant Ru-Ru scattering path (Fig. 4e)[38]. The coordination numbers and interatomic distances were determined from the EXAFS fitting results (Supplementary Fig. 30 and Table 11). Compared to the pristine Ru/TiO$_x$, the Ru-Ru interatomic distance slightly increased, while the Ru-O interatomic distance

slightly decreased, presumably due to the partial structural disorder of Ru/TiO$_x$ after OER[17]. The wavelet transforms (WT) of the Ru $K$-edge EXAFS oscillations show that the coordination features of Ru in Ru/TiO$_x$ are similar to those in Ru foil, with a predominant Ru-Ru coordination at 2.3 Å, indicating strong metallic coordination in the Ru/TiO$_x$ (Fig. 4g–j). Thus, the enriched electrons are transferred from TiO$_x$ to the empty $d$ bands of the confined Ru NPs, which stabilizes the coordination environment of Ru and prevents it from further oxidation.

The nanostructure integrity of an electrocatalyst at the atomic scale is also essential for achieving high acid OER performance. As shown in Supplementary Fig. 31, the nanorod-structure of Ru/TiO$_x$ was well maintained after both the 900 h and 50 h stability tests, which is attributed to the mechanical and chemical stability of the TiO$_x$ support. Furthermore, the nanoparticle size of Ru slightly increased to ~6 nm with uniform distribution due to the strong interaction between Ru and TiO$_x$ (Supplementary Fig. 32). In sharp contrast, for the annealed RuO$_x$/TiO$_2$, the TiO$_2$ nanorods were intertwined and connected with each other with the boundary blurred after long-time OER reaction, while the edge collapsed, which failed to stabilize and confine the size of Ru nanoparticles, resulting in the overgrowth and easy dissolution of RuO$_x$ active centers (Supplementary Figs. 33 and 34). In the commercial sample, the morphology collapsed and deactivated shortly after the OER process due to the weak binding of RuO$_2$ and TiO$_2$ by physical adsorption (Supplementary Fig. 33). Therefore, both the stable chemical states and well-maintained nanostructure contribute to the superior catalytic activity and long-term stability of Ru/TiO$_x$ electrocatalysts in acidic electrolyte.

## Mechanism analysis of OER activity

To further explore the possible catalytic mechanism on Ru/TiO$_x$, the pH-dependence measurements of the corresponding OER activities were performed. As a result, the Ru/TiO$_x$ shows pH-independent OER kinetics on the RHE scale, typical for AEM pathway (Supplementary Fig. 35). Density functional theory (DFT) calculations were performed to investigate the underlying mechanism of the superior OER performance of the Ru/TiO$_x$ catalyst. The Pourbaix diagrams were constructed to study the oxidation states of Ru and Ti under different pH and potential conditions (Supplementary Fig. 36). From the calculated Pourbaix diagrams, it can be seen that the stable oxidation states of Ti and Ru are TiO$_2$ and RuO$_2$ under the experimental potential condition, respectively. To obtain stable Ru/TiO$_x$ structures, we carried out ab initio molecular dynamics (AIMD) (Supplementary Figs. 37–40). The AIMD simulations were performed at 300 K for 10 ps with a time step 1 fs. These results provide the basis for our theoretical models. Our calculations focused on the Ru$_5$ cluster adsorbed on the TiO$_2$ (110) with one oxygen vacancy (V$_O$) sites as the selected model structure (Fig. 5a), which is marked as V$_{1O}$-Ru$_5$/TiO$_x$. By analyzing the results of Ru$_5$ adsorbed on substrates with different V$_O$ sites (Supplementary Fig. 39), we confirmed that the introduction of bridge V$_O$ contributes to the stabilization of the Ru$_5$ cluster by enhancing its adsorption strength. Specifically, when the Ru$_5$ cluster is adsorbed on the nearest neighbor site of the bridge V$_O$, the adsorption energy (E$_{ads}$) decreases significantly from −2.54 eV to −3.94 eV compared to Ru$_5$ adsorbed on pristine TiO$_2$ (P-Ru$_5$/TiO$_2$). Moreover, the average bond length of the Ru-Ru bonds is reduced to 2.50 Å after the introduction of the V$_O$ (Fig. 5b), which is consistent with the results of EXAFS analysis (Fig. 4e). The adsorption energy of oxygen ($\Delta E_O$) on the Ru sites can be a critical factor in determining the stability of the catalyst under acidic conditions[40]. As shown in Fig. 5b, the adsorption energy $\Delta E_O$ on the V$_{1O}$-Ru$_5$/TiO$_x$ is 0.73 eV higher than that of the P-Ru$_5$/TiO$_2$, indicating that the introduction of V$_O$ can improve the antioxidation ability of the Ru$_5$ cluster in the V-Ru$_5$/TiO$_x$. It has been reported that catalysts tend to pre-adsorb oxygen species and generate amorphous nonstoichiometric oxide layers under the OER working condition[13,41]. Therefore,

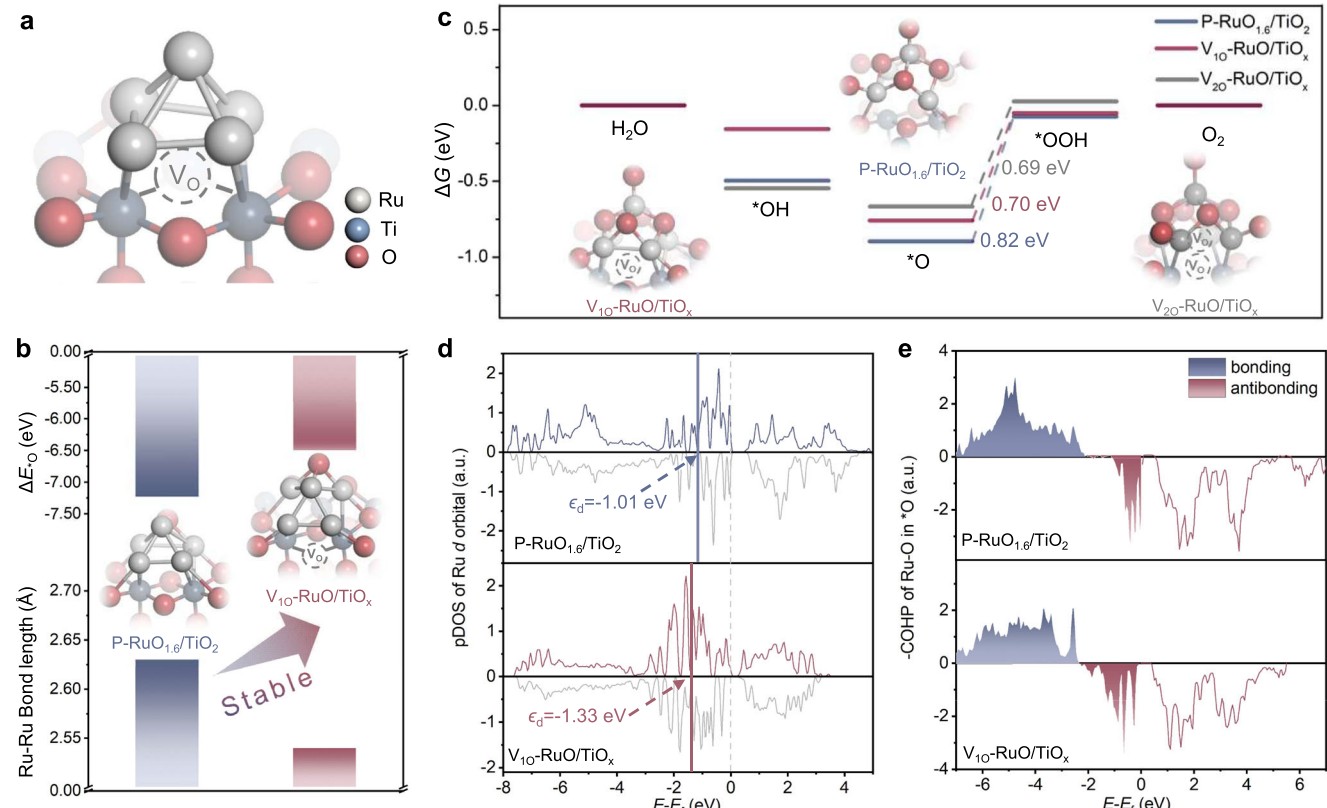

**Fig. 5 | Mechanism analysis of Ru/TiO_x towards acidic OER. a** Atomic structure of $Ru_5$ cluster adsorbed on the $TiO_2$ (110) surface with single oxygen vacancy ($V_{1O}$-$Ru_5$/$TiO_x$). **b** The adsorption energy of oxygen ($\Delta E_{*O}$) on the Ru sites and the corresponding Ru-Ru bond length value ($d_{Ru-Ru}$). The insets are the atomic structures of *O adsorbed on the structures of $Ru_5$ cluster adsorbed on the pristine $TiO_2$ (P-$Ru_5$/$TiO_2$) and $V_{1O}$-$Ru_5$/$TiO_x$. **c** Gibbs free energy profile of OER for P-$RuO_{1.6}$/$TiO_2$, $V_{1O}$-RuO/$TiO_x$ and $V_{2O}$-RuO/$TiO_x$ ($V_{2O}$ denotes 2 $V_O$s). The insets are the atomic structures of $V_{1O}$-RuO/$TiO_x$ and $V_{2O}$-RuO/$TiO_x$. **d** Projected density of states (PDOS) and band center of Ru $d$-state for P-$RuO_{1.6}$/$TiO_2$ and $V_{1O}$-RuO/$TiO_x$. **e** Crystal Orbital Hamilton Population (COHP) for the adsorbed *O of Ru-O in P-$RuO_{1.6}$/$TiO_2$ and $V_{1O}$-RuO/$TiO_x$.

the $Ru_5$ cluster will also be present as oxides during OER. Since the existence of oxygen vacancies will affect the oxidation state of Ru, we constructed different models ($V_{1O}$-RuO/$TiO_x$, $V_{2O}$-RuO/$TiO_x$, P-RuO/$TiO_2$ and P-$RuO_x$/$TiO_2$, $V_{2O}$ denotes two $V_O$) to study the OER process (Supplementary Fig. 40). Whether it is the initial $Ru_5$ cluster or $RuO_x$ after pre-oxidation, the introduction of $V_O$ helps to maintain the Ru atom at the interface in a lower valence state. Bader charge analysis shows that the charge of the Ru atom at the interface increases from 7.72 e⁻ in P-$Ru_5$/$TiO_2$ to 7.87 e⁻ and 8.02 e⁻ in $V_{1O}$-$Ru_5$/$TiO_x$ and $V_{2O}$-$Ru_5$/$TiO_x$, respectively. The results obtained for the pre-oxidized structures are in good agreement with this conclusion (Supplementary Fig. 41). The electron distribution is further proved by the charge density differences (Supplementary Fig. 42), the significant charge redistributions in $V_{1O}$-RuO/$TiO_x$ indicate stronger interaction between them than P-$RuO_{1.6}$/$TiO_x$.

In addition, the introduction of $V_O$ at the interface can not only enhance the antioxidant capacity of the catalyst, but also improve its performance for OER. Under acidic conditions, OER involves four proton-electron transfer steps at surface active sites, resulting in the formation of three intermediates (*OH, *O, and *OOH)[42,43]. The potential determining step (PDS) for all structural models evaluated is the third proton-electron transfer step of *O → *OOH, mainly due to the excessive adsorption of oxygen (Fig. 5c). Compared to P-$RuO_{1.6}$/$TiO_2$, the overpotential is reduced from 0.82 V to 0.70 V ($V_{1O}$-RuO/$TiO_x$) and 0.69 V ($V_{2O}$-RuO/$TiO_x$) when $V_O$ is introduced at the interface. The spin-polarized total density of states (TDOS) and projected density of states (PDOS) of Ru's $d$-orbitals for the P-$RuO_{1.6}$/$TiO_2$ and the V-RuO/$TiO_x$ models are shown in Fig. 5d and Supplementary Figs. 46 and 47.

The analysis of the $d$-band center results of Ru atoms at the interface shows a shift from −1.01 eV (P-$RuO_{1.6}$/$TiO_2$) to −1.33 eV ($V_{1O}$-RuO/$TiO_x$) and −1.66 eV ($V_{2O}$-RuO/$TiO_x$) upon the formation of $V_O$ (Fig. 5d and Supplementary Fig. 46), indicating the weaker binding to adsorbates according to the classical $d$-band theory[44]. More importantly, the V-RuO/$TiO_x$ systems have more states around Fermi level than P-RuO/$TiO_x$, indicating high electrical conductivity (Supplementary Fig. 47). It is possible that $V_O$ induces a stronger interaction at the interface and changes the electronic structure of the RuO/$TiO_x$ system. Furthermore, Crystal orbital Hamilton population (COHP) analysis was performed to evaluate the strength of Ru-O bonds in *O intermediates (Fig. 5e). The integrated area of COHP (ICOHP) is directly proportional to the bond strength, and a higher electron orbital overlap (lower ICOHP) indicates a stronger bond[45]. Compared to P-$RuO_{1.6}$/$TiO_2$, $V_{1O}$-RuO/$TiO_x$ exhibits increased occupancy of the anti-bonding state, resulting in a decrease in the -COHP of *O from 8.23 to 7.50, indicating a stronger Ru-O bond (Supplementary Table 12). Therefore, these results demonstrate the beneficial effect of introducing $V_O$ in maintaining the low valence state of the Ru active site and optimizing the binding energies of key intermediates, thereby significantly enhancing the activity and stability of OER.

## Discussion

In summary, we have proposed a one-step synthesis strategy by utilizing the redox reaction between Ru ions and Ti substrate to construct non-stoichiometric $TiO_x$ supported Ru NPs as a binder-free electrode towards OER in acidic media. The as-prepared Ru/$TiO_x$ catalyst demonstrates both record high activity and high stability: the

overpotentials are only 174 and 209 mV to achieve current densities of 10 mA cm$^{-2}$ and 100 mA cm$^{-2}$, respectively, with a stability for 900 h under 10 mA cm$^{-2}$. This enhanced performance also allows the Ru/TiO$_x$ to work in pure natural seawater electrolysis. Both experimental results and DFT calculations revealed the crucial role of in situ non-stoichiometric oxides, which induced structural confinement and charge accumulation at Ru sites, preventing the aggregation and over-oxidation of Ru, thereby maintaining the OER performance during the long-term stability test. The concepts of in situ metal-support inter-action and fabrication of nonstoichiometric compounds not only offer paths to the next-generation OER catalysts but also illustrate a pro-mising way to design active sites for nanocatalysts across the wide range of conceivable systems.

## Methods

### Synthesis of Ru/TiO$_x$ samples

Ru/TiO$_x$ self-supporting electrode was prepared through a one-step hydrothermal method. Prior to the synthesis, a piece of Ti foam (1.0 × 6.0 cm, thickness of 0.6 mm) was sonicated in distilled water, acetone, and ethanol sequentially to remove surface oil stains and oxide layer, followed by etching the TF in the solution of HCl (18 wt%) at 363 K for 15 min. In a typical procedure, a certain amount of RuCl$_3$·3H$_2$O was dissolved in deionized water (10 mmol·L$^{-1}$) with HCl (3 wt%) under vigorous stirring to form a uniform solution. The resulting solution was transferred into a 50 mL Teflon-lined stainless-steel autoclave with a piece of etched TF immersed into the reaction solution. The autoclave was sealed and heated to 200 °C for 20 h, and then cooled to room temperature. The obtained material was washed three times with deionized water in ultrasound (10 s each) and then treated by vacuum drying at 60 °C overnight.

### Synthesis of control samples

For comparison, an in situ formed bare TiO$_2$/TF was synthesized using the etched-TF via the same method without the addition of RuCl$_3$·3H$_2$O. The obtained Ru/TiO$_x$ was further annealed in air at 450 °C for 1 h to prepare annealed RuO$_x$/TiO$_2$. The benchmark IrO$_2$ and RuO$_2$ catalysts on Ti foam were fabricated by the following steps: 280 μL ethanol, 20 μL Nafion, 70 μL deionized water, and 50 mg IrO$_2$/RuO$_2$ (Alfa Aesar) were mixed to prepare a dispersion, then add the prepared dispersion to the TF (the mass added to the electrode is determined according to the loading mass).

### Material characterizations

The morphology, energy dispersion spectra (EDS), and elemental mapping of the samples were collected by a scanning electron microscope (FE-SEM, JSM-7500, Japan), transmission electron micro-scope (FE-TEM, G2F20, USA) and scanning transmission electron microscopy (STEM, Tecnai G2F30). Powder X-ray diffraction (XRD) patterns were collected using a Rigaku Smartlab diffractometer with Cu Kα radiation (λ = 1.5418 Å). The elements and corresponding valence states were analyzed by X-ray photoelectron spectroscopy (XPS) (ESCLAB 250Xi, Thermo Fisher Scientific). Aberration-corrected high angle annular dark field-scanning TEM (HAADF-STEM) images were collected using a Titan Cubed Themis G2 300 Double Aberration-Corrected Transmission Electron Microscope operating at 300 kV. The liquid and bubble contact angles were measured by the captive bubble method at ambient temperature (Dataphysics-OCA50, German). The accurate contents of Ru element in electrolytes were characterized by the inductively coupled plasma mass spectrometry (ICP-MS) mea-surement (iCAP Q, Thermo, Waltham, USA). The synchrotron-based X-ray absorption fine structure (XAFS) measurements were performed with Si (311) crystal monochromators at the BL14W1 beamlines at the Beijing Synchrotron Radiation Facility (BSRF) (Beijing, China). The XAFS spectra were recorded at room temperature using a 4-channel Silicon Drift Detector (SDD) Bruker 5040. Ru K-edge extended X-ray

absorption fine structure (EXAFS) spectra were recorded in transmis-sion mode. Negligible changes in the line-shape and peak position of Ru K-edge XANES spectra were observed between two scans taken for a specific sample. The XAFS spectra of the standard samples (Ru foil and RuO$_2$) were recorded in transmission mode. The spectra were processed and analyzed by the software codes Athena and Artemis. In-situ XPS spectra were measured by ambient pressure XPS end station equipped with a static electrochemical cell at ESCLAB 250Xi (Supple-mentary Fig. 27). The counter electrode was Pt and the reference electrode was saturated calomel electrode (SCE). The potentials of the Ru/TiO$_x$ as a working electrode (1.0−1.7 V vs. RHE) were precisely controlled.

### Electrochemical measurements

Electrochemical measurements were performed with a workstation in a typical three-electrode configuration consisting of a Pt plate (the counter electrode), an Ag/AgCl electrode (the reference electrode), and the active material (the working electrode) in 0.5 M H$_2$SO$_4$ solution (pH = 0.3). The pH of the electrolyte was measured using a pH-meter (Mettler Toledo, Germany). The linear sweep voltammetry (LSV) polarization curves were collected at a scan rate of 5 mV s$^{-1}$. Therein, all of the measured potentials versus the reference electrode were con-verted to a reversible hydrogen electrode (RHE) according to the equation ($E_{RHE} = E_{Ag/AgCl} + 0.197 V + 0.0591 × pH$). The overpotential values reported in the manuscript are all obtained through the LSV curves without iR correction. For comparison, the LSV curves with 95% iR (i, current; R, resistance) compensation were also reported. The OER stability was evaluated by continuous cyclic voltammograms (CVs) up to 550 mA cm$^{-2}$ for 50 cycles and chronopotentiometry test at con-stant current densities of 10 mA cm$^{-2}$ and 100 mA cm$^{-2}$. The chron-opotentiometric tests of the samples under a constant OER current density of 10 and 100 mA cm$^{-2}$ were conducted in an H-type water electrolysis cell with the anode and cathode separated by a Nafion 117 membrane. The LSV polarization curves and chronopotentiometric results were obtained under the same operation conditions in natural seawater (Yellow Sea, 120°E, 35°N) without any further treatment. Nyquist plots of electrochemical impedance spectroscopy (EIS) mea-surements were collected in the frequency range of 100 kHz to 0.01 Hz at open-circuit potential with an amplitude of 5 mV AC voltage in 0.5 M H$_2$SO$_4$ solution. The electrochemical accessible surface area (ECSA) was determined by: ECSA = $C_{dl}/C_s$, where $C_{dl}$ is double-layer capaci-tance and $C_s$ the specific capacitance of the sample. In this work, a general specific capacitance of $C_s$ = 0.035 mF cm$^{-2}$ was used based on typically reported values[3,28]. $C_{dl}$ was determined by the equation $C_{dl} = i_c/v$, where $i_c$ is the charging current and $v$ is the scan rate. A series of CV tests in the non-faradaic potential region (0.2 - 0.3 V vs. RHE) with various scan rates (20, 40, 60, 80, 100, 120 mV s$^{-1}$) were performed. By plotting measured $i_c$ versus $v$, $C_{dl}$ was obtained from the slopes of the linear fitting. Typically, the pH-dependence measurement was carried out at 1.23 to 1.53 V vs. RHE in H$_2$SO$_4$ with different pH (0.3, 0.4, 0.7, and 1). For electrolyser tests, a self-made cell was used as the PEMWE device and a cation exchange membrane (Nafion 212) as the membrane electrolyte. The membrane electrode assembly (MEA) was prepared by pressing the cathodes (20% Pt/C sprayed on the Nafions 212 membrane) and anodes (self-supported Ru/TiO$_x$). During the test, the cell was maintained at 60°C, and the pre-heated deionized water was fed to the anode by a peristaltic pump. All the data of PEMWE were not iR corrected and displayed as raw data.

### Computational details

Density functional theory (DFT) calculations[46,47], were performed as implemented in the Vienna ab initio simulation package (VASP 5.4.4)[48,49] in conjunction with VASPsol[50,51]. The projected augmented wave (PAW) potential and the generalized gradient approximation (GGA) with the Perdew-Burke-Ernzerhof (PBE) functional were used to

describe the electron-ion interactions and the exchange-correlation energy, respectively. The cutoff energy for the plane-wave basis set was set to 500 eV. A force tolerance was set as 0.02 eV Å$^{-1}$ on each atom for structural relaxation. A $2 \times 3 \times 1$ Monkhorst-Pack $k$-point sampling was used for geometry optimization, and a $3 \times 4 \times 1$ Monkhorst-Pack $k$-point sampling in the Brillouin zone was used for electronic structure calculation. A Hubbard correction of $U = 2.5$ eV, $J = 0$ eV, and $U = 3.0$ eV, $J = 0.5$ eV was applied to Ti and Ru atoms. The (110) surface of $TiO_2$ was simulated as a four-layer slab with ~15 Å vacuum layer along the c-axis, where the bottom $TiO_2$ were fixed at their optimal bulk positions and the remaining atoms were fully relaxed.

In this work, we considered OER to occur as the following four steps:

$$* + OH^- \rightarrow *OH + e^- \tag{1}$$

$$*OH + OH^- \rightarrow *O + H_2O + e^- \tag{2}$$

$$*O + OH^- \rightarrow *OOH + e^- \tag{3}$$

$$*OOH + OH^- \rightarrow * + O_2(g) + H_2O + e^- \tag{4}$$

where * is the adsorption site for the intermediates. For each step of OER, the free energy of reaction $\Delta G$ was studied within the framework of the computational hydrogen electrode[43,52], where $H_2$ is in equilibrium with both protons and electrons:

$$\frac{1}{2}H_2 \rightarrow H^+ + e^- \tag{5}$$

Then $\Delta G$ was calculated by the following equation:

$$\Delta G = \Delta E + \Delta ZPE - T\Delta S + \Delta G_U \tag{6}$$

where $\Delta E$ is the DFT energy difference of the reactions, $\Delta ZPE$ is the zero-point energy correction, $\Delta S$ is the vibrational entropy change at 298.15 K, $\Delta G_U = -eU$, $U = 1.23$ V.

The overpotential $\eta$ of OER can be evaluated from the largest $\Delta G$ of each step as:

$$\eta_{OER} = \frac{\max\{\Delta G_1, \Delta G_2, \Delta G_3, \Delta G_4\}}{e} \tag{7}$$

where $\Delta G_1, \Delta G_2, \Delta G_3,$ and $\Delta G_4$ are the free energy of reactions (1) to (4), respectively.

Ru NP is simulated by $Ru_5$ clusters with the configuration of the tetragonal cone, and non-stoichiometric oxide $TiO_x$ is simulated by $TiO_2$ (110) with O vacancy ($V_O$) on the surface (V-$Ru_5$/$TiO_x$), while $Ru_5$ clusters adsorbed on the pristine $TiO_2$ (110) (P-$Ru_5$/$TiO_2$) were also investigated for comparison. Considering that $V_O$ could enhance the antioxidant capacity of $Ru_5$ cluster, P-$RuO_{1.6}$/$TiO_2$ and V-$RuO$/$TiO_x$ were constructed to investigate the OER mechanism in the catalytic performance calculations. Detailed structural information is presented in the supplement materials. To obtain a stable $TiO_x$ surface, we carried out ab initio molecular dynamics (AIMD). The AIMD simulations at 300 K for 10 ps with a time step 1 fs. The Charge density differences ($\Delta\rho = \rho_{RuO/TiOx} - \rho_{RuO} - \rho_{TiOx}$) are calculated to express the interaction between RuO and $TiO_x$. For V-$RuO$/$TiO_x$, the significant charge redistributions indicate a stronger interaction between them than P-$RuO$/$TiO_x$.

## Data availability

The data that support the findings of this study are presented in the main text and the Supplementary Information, and are available from the corresponding authors upon reasonable request. The Source Data underlying the figures of this study are available at https://doi.org/10.6084/m9.figshare.24324541. All raw data generated during the current study are available from the corresponding authors upon request.

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

## Acknowledgements

This work was sponsored by Tsinghua University-Toyota Joint Research Center for Hydrogen Energy and Fuel Cell Technology of Vehicles.

## Author contributions

R.T.L. and L.X.Z. conceived the original idea. R.T.L. supervised the project. L.X.Z. carried out catalyst synthesis, materials characterization, catalytic tests, and related data processing. Y.-F.S., F.Y. and J.L. carried out DFT calculations. L.X.Z. and Y.F.S. co-wrote the manuscript. L.X.Z., Y.F.S., F.Y.K., R.T.L. and J.L. performed the analysis and revised the manuscript. All authors contributed to data analysis, and scientific discussion and commented on the manuscript.

## Competing interests

The authors declare no competing interests.
