## [Peer review file · Nature Communications]

REVIEWER COMMENTS

Reviewer #1 (Remarks to the Author):

Developing high-performance OER catalysts for sustainable hydrogen production by water splitting in acidic media is highly desired but remains a big challenge. The authors report a non-stoichiometric TiO_x-supported Ru as a binder-free electrode, which achieves high current densities under low overpotentials with a record high stability for operating over 37 days. The strategy proposed by the author is very interesting, and the catalysts have been extensively investigated both experimentally and theoretically. I would be pleased to recommend this manuscript to be published in Nature Communications once the following comments could be well addressed.

1. In the process of one-step growth of Ru/TiO_x, how to reduce Ru while growing TiO_x? Please further explain the growth mechanism of the material.
2. The intrinsic activities of Ru/TiO_x should be further assessed based on the turnover frequencies (TOFs) at different overpotentials. The TOF comparison of the as-prepared Ru/TiO_x and commercial RuO₂ catalyst should be considered.
3. The authors have emphasized the charge transfer between TiO_x and Ru is the key to maintain the OER activity of Ru. It is recommended to add the analysis of charge density difference in the DFT calculation.
4. The author adopted the AEM mechanism to calculate the OER path. It is recommended to rule out the LOM mechanism through pH-dependent experiments.
5. How does the valence state of Ti change during the OER process? At present, the authors analyzed the valence state changes of Ru before and after OER. It is recommended to add the valence change analysis of the substrate as well.
6. Ru/TiO_x can achieve high current density of 500 mA cm⁻² under overpotential of 265 mV. Compared with other acidic OER catalysts, what is the main reason for its ability to achieve high current density?

Reviewer #2 (Remarks to the Author):

This paper presents an investigation into a method aimed at stabilizing Ru-based catalysts for acidic water oxidation reactions through the use of non-stoichiometric titanium oxides (TiO_x). A significant improvement in the stability and activity of the Ru/TiO_x catalyst, compared to traditional stoichiometric TiO₂ supports, is reported by the authors. Notably, the synthesis approach is streamlined, involving a single-step process in which Ti foam is employed as a support for the catalyst. This work represents a valuable contribution to the quest for developing cost-effective and stable catalysts for acidic water oxidation.

However, the paper falls short in a few key areas, most notably in the depth and rigor of characterization and system-related analyses. Additionally, the computational calculations presented in the study appear overly simplified and, as such, insufficient to robustly support the experimental results. The specific concerns are elaborated upon as follows:

1. Unconvincing Statement on Non-Stoichiometric TiO₂ and Ru Oxidation State:

The current assertions regarding the non-stoichiometric nature of TiO₂ and the oxidation state of Ru under reaction conditions are not compelling. The authors should perform in-situ X-ray Absorption Spectroscopy (XAS) at the Ti and Ru K-edges. This would enable a deeper and more precise understanding of the oxidation states and chemical environments of Ti and Ru under actual reaction conditions, and potentially strengthen the argument presented.

2. Necessity for Real-World Testing Conditions:

The long stability tests presented in this work are insufficiently related to practical, industrial scenarios. The authors should conduct stability tests under conditions that are representative of industrial applications, specifically within a real membrane electrode assembly (MEA) that includes a proton exchange membrane, and at industrially relevant current densities. The use of 10 mA/cm² for testing seems to fall short of the conditions that this material would face in a real-world application.

3. Need for Pourbaix Diagram in Computational Study:

The computational chemistry aspects of the work could be enhanced significantly. The authors should utilize the Pourbaix diagram in their computational study. This would help to understand the thermodynamic stability of the various possible oxidation states of Ru and Ti under different pH and potential conditions, and could elucidate the origin of the observed high stability of the material.

4. Simplistic Model of Non-Stoichiometric TiO₂:

The non-stoichiometric TiO₂ model with a single oxygen vacancy, as used in this study, appears overly simplistic. The authors should consider adopting a more complex and realistic model of non-stoichiometric TiO₂. Optimization using molecular dynamics simulations may offer a more accurate representation of the actual material and its behavior under operational conditions.

5. Comparison of Experimental and Theoretical Turnover Frequency (TOF):

While the geometry current density improvement of Ru/TiO_x is noteworthy, the ECSA (Electrochemical Surface Area) normalized improvement does not appear to be significant. Given that reaction energies have been calculated in the study, the authors should also calculate the theoretical turnover frequency (TOF) and compare it with the experimental TOF. This would give a more comprehensive understanding of the catalytic efficiency of the material system under investigation.

6. Discrepancy Between Electrical Conductivity and DFT Calculations:

The authors mention that the "substoichiometric phase Ti₂O₃ exhibits high electrical conductivity". However, their DFT calculations suggest the TiO_x material is a semiconductor with a large band gap. The authors should resolve this apparent contradiction. A comprehensive explanation is necessary to clarify this discrepancy and align the DFT calculations with the claimed electrical properties of the material.

Reviewer #3 (Remarks to the Author):

The authors report a Ti oxides (TiO_x) supported Ru electrocatalyst (Ru/TiO_x) working in an acidic electrolyte. The Ru/TiO_x exhibits low OER overpotentials of 174, 209, and 265 mV to reach current densities of 10, 100, and 500 mA cm⁻², respectively, and is stable for 900 h under 10 mA cm⁻² in 0.5 M H₂SO₄. Their theoretical calculations indicate that oxygen vacancy (VO) defects at the Ru-TiO_x interface induced a charge accumulation at Ru sites, thereby preventing the aggregation and over-oxidation of Ru during the OER process. It is, generally, an interesting work. However, the overall novelty of the work is not sufficient for publication in Nature Communications. In fact, many reports have already shown an efficient way to improve the acidic OER stabilities of Ru-based materials by introducing electron-donated supports to prevent the over-oxidation of Ru sites during the reaction (Nature Catalysis 2019, 2, 304-313; Angew. Chem. Int. Ed. 2022, 61, e202202519). The effectiveness of this strategy has also been confirmed in VO defective TiO₂ supported RuO₂ catalyst (ACS Catal. 2022, 12, 9437-9445). The originality of the work is somehow limited as previous Ru/TiO₂ has been made. Moreover, from catalytic performance perspective, the activity and stability of Ru/TiO_x for acidic OER are not particularly better than those recently reported Ru-based catalysts (Nature Commun. 2023, 14, 843/1412/2517; Nat. Mater. 2023, 22, 100-108).

Some other comments for authors to consider:

1. The potential of reference electrode, Ag/AgCl, needs to be calibrated by a reversible hydrogen electrode (RHE) at first, according to which the reported OER potentials on the RHE scale will be more credible. As well-known, the pH value of 0.5 M H₂SO₄ is around 0.3 rather than 0.0. In the method section, authors do not report the pH value of electrolyte used for the potential conversion based on

the equation (Line 511, page 24).

2. Also for the reported potentials, are the values already iR corrected? Please, check it and if they are iR corrected change the axis to E-iR and report details for iR correction in the method section and how much is corrected. The results change a lot if the graphs are corrected or not. Please provide non-iR corrected data for Fig. 2a in the SI.

3. The observed low overpotential may also be due to the partial oxidation of water to hydrogen peroxide in addition to the O₂ evolution (Sci Bull, 2023, 68, 613-621; J. Phys. Chem. Lett. 2015, 6, 4224-4228). The authors need to prove more convincing data to support their claims.

4. The Tafel plots should be derived from the steady-state polarization curves (ACS Energy Lett. 2021, 6, 1607).

5. The applied potentials should be reported for the EIS measurements in Fig. 2c.

6. The OER durability of Ru/TiO_x needs to be further checked under the cycling condition (at least up to 200 mA cm⁻²). Could the authors measure more than 50 OER cycles up to 500 mA cm⁻²?

7. Is the stability test under constant current density also performed in an undivided cell? Please check it. Because the dissolved Ru cations from the catalyst will be readily reduced and redeposited on the counter electrode in an undivided cell. Thus, if an undivided cell is used, the detected Ru ions content in electrolyte after the stability test shown in Fig. 3d (upper plot) cannot be trusted.

8. Considering that XPS probes the top few nanometers within a micron region of materials, the change of ruthenium content determined by XPS before and after the OER test is meaningless (Fig. 3d, lower plot).

9. The deconvolution of XPS spectra in Figs. 2a-c is very slipshod. There lacks consistent full width at half maximum and spin-orbit splitting of Ru 3d. The C 1s spectrum at 284.8 eV overlapped with Ru 3d_{3/2} is also not reported. Please use a supplementary table to summarize the fit parameters of XPS spectra.

10. In general, higher oxidation state of Ru-sites leads to better activity (J. Phys. Chem. C 2017, 121, 18516-18524; ACS Catal. 2020, 10, 12182-12196). The lower the oxidation state of Ru will tend to result in stronger OH* binding and higher overpotentials. This is in stark contrast to what the authors find. The possible OER mechanism needs to be discussed among many mechanisms suggested in recent literatures.

11. In Figs. 5d-e, the DFT results are reported on the base of two O-adsorbed structures, V-RuO/TiO_x and P-RuO_{1.6}/TiO₂, rather than the pristine Ru/TiO₂ and Ru/TiO_x. It is unclear why the authors treated it this way. The reasons should be provided.

Point-by-point response to the reviewers' comments

Reviewer #1 (Remarks to the Author):

Developing high-performance OER catalysts for sustainable hydrogen production by water splitting in acidic media is highly desired but remains a big challenge. The authors report a non-stoichiometric TiO_x-supported Ru as a binder-free electrode, which achieves high current densities under low overpotentials with a record high stability for operating over 37 days. The strategy proposed by the author is very interesting, and the catalysts have been extensively investigated both experimentally and theoretically. I would be pleased to recommend this manuscript to be published in *Nature Communications* once the following comments could be well addressed.

Comment #1-1) In the process of one-step growth of Ru/TiO_x, how to reduce Ru while growing TiO_x? Please further explain the growth mechanism of the material.

Response: We appreciate the reviewer's constructive comments very much. The principle of one-step growth of the Ru/TiO_x catalyst is based on the redox interaction-engaged strategy which is driven by thermodynamics (*Chem. Soc. Rev.* 2020, **49**, 736-764). In detail, if the difference in standard electrode potentials is higher than zero, the reaction will proceed thermodynamically. **Table R1** summarizes the standard reduction potentials of Ru and Ti species at 25 °C under 1 atm (*Handbook of Chemistry and Physics*, 84th ed., CRC Press, Boca Raton, FL 2004). Although specific synthesis reactions (200 °C in autoclave) were performed under nonstandard conditions, potential data can be applied as a basic guide when choosing potential redox pairs. On this basis, in our work, the Ti foam serves as both substrate and reducing agent to efficiently reduce Ru³⁺ ions through a redox reaction due to a thermodynamically favorable process according to the standard potentials of different electrode pairs shown in **Table R1**. The following description has been added to the revised manuscript (Page 5) as below:

On page 5: "Through a facile hydrothermal process, the in situ growth of TiO_x and the loading of Ru nanoparticles (NPs) are realized in one step. Specifically, commercial Ti foam (TF) with porous structure and good chemical/mechanical stability is selected as both substrate and Ti source for TiO_x growth. The Ti foam serves as both substrate and reducing agent to efficiently reduce Ru³⁺ ions through redox interaction-engaged strategy due to a thermodynamically favorable process."

Table R1. The standard potentials of different metal ions^{1,2}.

Reaction	Potential values (V)
$Ru^{3+} + e^{-} \rightleftharpoons Ru^{2+}$	0.249
$Ru^{2+} + e^{-} \rightleftharpoons Ru$	0.455
$TiO(s) + 2H^{+} + 2e^{-} \rightleftharpoons Ti(s) + H_2O$	-1.31
$TiO_2(s) + 4H^{+} + 4e^{-} \rightleftharpoons Ti(s) + 2H_2O$	-0.86
$Ti_2O_3(s) + 2H^{+} + 2e^{-} \rightleftharpoons 2TiO(s) + H_2O$	-1.23

References

1. Wang, X., Song, S. & Zhang, H.-J. A redox interaction-engaged strategy for multicomponent

2. *Handbook of Chemistry and Physics*, 84th ed., CRC Press, Boca Raton, FL 2004

Comment #1-2) The intrinsic activities of Ru/TiO_x should be further assessed based on the turnover frequencies (TOFs) at different overpotentials. The TOF comparison of the as-prepared Ru/TiO_x and commercial RuO₂ catalyst should be considered.

Response: We thank the reviewer very much for the constructive suggestions. To further elucidate the intrinsic mechanism for the enhancement of OER activity, per-site turnover frequency (TOF) is employed to compare the practical performance of catalysts. As a result, the TOF of Ru/TiO_x is calculated to be 1.960 s⁻¹ at an overpotential of 300 mV, which is 5.13 times higher than that of RuO₂ and superior to representative OER catalysts in various acidic media. The corresponding discussion of TOF has been added to the revised manuscript (Page 10) and the Supplementary Information (*Supplementary Fig. 14 and Tables 4-5*, Page 5, 12 and 35) as below:

On page 10: “The intrinsic activities of Ru/TiO_x were further assessed based on turnover frequencies (TOFs) at different overpotentials, which are among the highest when compared with representative OER catalysts in various acidic media (*Supplementary Fig. 14a and Table 4*). For example, the TOF of Ru/TiO_x is calculated to be 1.960 s⁻¹ at an overpotential of 300 mV based on the total loading mass, which increases to 2.192 s⁻¹ when calculated based on ECSA (*Supplementary Table 5*). We also provided a bar graph of ECSA-normalized current densities and TOF values at an overpotential of 300 mV, which shows similar activity trend (*Supplementary Fig. 14b*). The above results prove that Ru/TiO_x shows the highest intrinsic activity per site.”

On SI page 5: “**Calculation of turnover frequency (TOF):** The TOF value based on inductively coupled plasma mass spectrometry (ICP-MS) results (bulk TOF) and electrochemical active surface area (ECSA) values (ECSA TOF) were calculated and compared. The Bulk TOF value was calculated by following formula:

$$\text{TOF} = \frac{\# \text{ Total Oxygen Turn Overs per geometric area}}{\# \text{ Active sites per geometric area}} \quad (1)$$

Total oxygen turn overs per geometric area

$$\begin{aligned} &= \left(j \frac{\text{mA}}{\text{cm}^2} \right) \left(\frac{1 \text{ C s}^{-1}}{1000 \text{ mA}} \right) \left(\frac{1 \text{ mol O}_2}{4 \text{ mol e}^-} \right) \left(\frac{6.022 \times 10^{23} \text{ O}_2 \text{ atoms}}{1 \text{ mol O}_2} \right) \left(\frac{1 \text{ mol e}^-}{96485.3 \text{ C}} \right) \\ &= 1.56 \times 10^{15} \left(\frac{\text{O}_2 \text{ s}^{-1}}{\text{cm}^2} \right) \text{ per } \left(\frac{\text{mA}}{\text{cm}^2} \right) \end{aligned} \quad (2)$$

Active sites per geometric area

$$= \left(\frac{\text{mass loading of Ru } (\text{g}/\text{cm}^2)}{\text{Ru Mw } (\text{g}/\text{mol})} \right) \left(\frac{6.022 \times 10^{23} \text{ Ru atoms}}{1 \text{ mol Ru}} \right) \quad (3)$$

The ECSA TOF value was calculated by Equation (2) and the following formula:

$$\text{TOF} = \frac{\# \text{ Total Oxygen Turn Overs per geometric area}}{\# \text{ Active sites per real area}} \quad (4)$$

Active sites per real surface area:

$$= \left(\frac{\text{number of active sites / unit cell}}{\text{unit cell volume}} \right)^{2/3} \quad (5) \quad ''$$

On SI page 12:

Supplementary Figure 14. a, Potential-dependent turnover frequency (TOF) curves. **b**, The corresponding bar graph of the ECSA-normalized current density (blue) and TOF values (red) at the overpotential of 300 mV.

On SI page 35:

Supplementary Table 4. TOF of Ru/TiO_x with previously reported OER catalysts in acid.

Catalysts	Electrolyte	Overpotential (mV)	TOF (s ⁻¹)	Reference
Ru/TiO _x	0.5 M H ₂ SO ₄	270	1.707	This work
		300	1.960	
Annealed RuO _x /TiO ₂	0.5 M H ₂ SO ₄	300	0.820	This work
Com. RuO ₂	0.5 M H ₂ SO ₄	300	0.322	This work
SnRuO _x	0.5 M H ₂ SO ₄	250	0.63	Nat. Commun. 14 , 843 (2023)
Rh-RuO ₂ /Graphene	0.5 M H ₂ SO ₄	300	1.74	Nat. Commun. 14 , 1412 (2023)
high-loading Ir single atoms with d -band holes	0.1 M HClO ₄	216	0.599	Angew. Chem., Int. Ed. 135 , 202308082 (2023)
Ru ₅ W ₁ O _x	0.5 M H ₂ SO ₄	300	0.163	Nat. Commun. 13 , 4871 (2022)
Cr-SrIrO ₃	0.1 M HClO ₄	300	0.208	Nano Energy 102 , 107680 (2022)
Ru/MnO ₂	0.1 M HClO ₄	165	0.331	Nat. Catal. 4 , 1012-1023 (2021)
Ru ₁ Ir ₁ O _x	0.5 M H ₂ SO ₄	300	0.47	Adv. Energy Mater. 11 , 2102883 (2021)

Supplementary Table 5. TOF of catalysts using different normalization methods.

Catalysts	Overpotential (mV)	Bulk TOF (s ⁻¹)	ECSA TOF (s ⁻¹)
Ru/TiO _x	270	1.707	1.835
	300	1.960	2.192
Annealed RuO _x /TiO ₂	300	0.820	1.640
Com. RuO ₂	300	0.322	1.520

Comment #1-3) The authors have emphasized the charge transfer between TiO_x and Ru is the key to maintain the OER activity of Ru. It is recommended to add the analysis of charge density difference in the DFT calculation.

Response: We appreciate for the important comments. In the previous version of our manuscript, we used the Bader charge analysis to demonstrate the electron transfer between Ru and TiO_x support. Following the reviewer's suggestion, we have supplemented the analysis of differential charge density to visually demonstrate the charge-redistribution. The following description has been added to the revised manuscript (Page 20) and Supplementary Information (Page 29) as below:

On page 20: “The electron distribution are further proved by the charge density differences (Supplementary Fig. 42), the significant charge redistributions in $V_{10}\text{-RuO}/\text{TiO}_x$ indicate stronger interaction between them than $P\text{-RuO}_{1.6}/\text{TiO}_x$.”

On SI page 29:

Supplementary Figure 42. Side view and top view of the differential charge density of $V_{10}\text{-RuO}/\text{TiO}_x$ (a,b) and $P\text{-RuO}_{1.6}/\text{TiO}_2$ (c,d). Electron accumulation and depletion are shown in cyan and yellow, respectively. (isovalue of $0.01|e|/\text{Bohr}^3$)

Comment #1-4) The author adopted the AEM mechanism to calculate the OER path. It is recommended to rule out the LOM mechanism through pH-dependent experiments.

Response: Thanks for the important suggestions. Determining the OER path is crucial for understanding OER mechanisms. Since the adsorbate evolution mechanism (AEM) pathway involves four electron-proton transfer steps, it is characterized by pH-independent activity on the reversible hydrogen electrode (RHE) scale (*Adv. Mater.* 10.1002/adma.202305939; *Adv. Mater.* 2023, **35**, 2210565; *ACS Catal.* 2023, **13**, 256-266; *Energy Environ. Sci.* 2022, **15**, 2356). In contrast, the lattice oxygen-evolution mechanism (LOM) pathway involves non-concerted proton-electron transfers and therefore exhibits pH-dependent activity (*Nat. Chem.* 2017, **9**, 457-465; *Nat. Commun.* 2022, **13**, 4871). To further explore the possible catalytic mechanism on Ru/ TiO_x , the pH-dependence measurements of the corresponding OER activities were

performed in the pH range of 0.3-1. As a result, the Ru/TiO_x shows pH-independent OER kinetics on the RHE scale, which is typical for AEM pathway and consistent with the mechanism analysis in our original manuscript. The following description has been added to the revised manuscript (Page 18 and 25) and Supplementary Information (*Supplementary Fig. 35*, Page 25) as below:

On page 18: “To further explore the possible catalytic mechanism on Ru/TiO_x, the pH-dependence measurements of the corresponding OER activities were performed. As a result, the Ru/TiO_x shows pH-independent OER kinetics on the RHE scale, typical for AEM pathway (*Supplementary Fig. 35*).”

On page 25: “Typically, the pH-dependence measurement was carried out at 1.23 to 1.53 V vs. RHE in H₂SO₄ with different pH (0.3, 0.4, 0.7 and 1).”

On SI page 25:

Supplementary Figure 35. a, OER activity of Ru/TiO_x with varying pH. b, pH dependence on the OER potential at different current densities for Ru/TiO_x.

Comment #1-5) How does the valence state of Ti change during the OER process? At present, the authors analyzed the valence state changes of Ru before and after OER. It is recommended to add the valence change analysis of the substrate as well.

Response: We appreciate the important comment of the reviewer very much. We fully agree that it is very crucial to study the oxidation state changes of both Ru and Ti under actual reaction conditions. Therefore, we have conducted *in-situ* XPS to study the changes in the oxidation states of Ru and Ti in the catalyst. Through fitting and analysis of *in-situ* XPS data, we compared the peak position changes and the proportion of each valence state of Ru and Ti in detail. As a result, the oxidation state of Ru remains unchanged, while the ratio of Ti³⁺/Ti⁴⁺ and O_v/O_L (oxygen vacancy/lattice oxygen) slightly changes and remains stable, indicating the critical role of the Ti³⁺ and O_v in stabilizing Ru active sites. The overall conclusion is in good agreement with the *ex-situ* XPS analysis. The following discussion has been added to the revised manuscript (Page 16 and 24) and Supplementary Information (*Supplementary Figs. 27-29* and *Table 9*, Page 19 and 39) as below:

On page 16: “Afterwards, the chemical environments of Ti and Ru in the Ru/TiO_x catalyst under actual OER conditions were monitored by *in-situ* XPS (*Supplementary Figs. 27-29*). Significantly, the Ru 3d XPS

peaks at 280.1 and 284.2 eV exhibit negligible changes with the applied potential increased from 1.0 to 1.7 V vs. RHE (Supplementary Fig. 28a and Table 9). Detailed quantitative analysis shows the ratio of $\text{Ru}^{n+}/\text{Ru}^0$ remains almost identical at 0.7 as the voltage increases, which is consistent with the ex-situ XPS analysis results (Supplementary Figs. 26 and 29). The above results further confirm the stable Ru chemical state in Ru/TiO_x during the OER process. For the Ti 2p spectra (Supplementary Fig. 28b and Table 9), the coexistence of Ti^{3+} and Ti^{4+} species was distinguished. Interestingly, as the voltage increases, the ratio of $\text{Ti}^{3+}/\text{Ti}^{4+}$ slightly decreases and remains stable at 0.17, indicating the critical role of the Ti^{3+} in stabilizing Ru active sites. Corresponding to this phenomenon is the O_V/O_L (oxygen vacancy/lattice oxygen) ratio obtained from the O 1s spectra (Supplementary Figs. 28c and 29). It can be seen that as the voltage increases, the O_V/O_L ratio first decreases and then almost returns to the initial state, manifesting that O_V regeneration is accompanied by the release of oxygen, thereby stabilizing active species³⁹.”

On page 24: “In-situ XPS spectra were measured by ambient pressure XPS end station equipped with a static electrochemical cell at ESCLAB 250Xi (Supplementary Fig. 27). The counter electrode was Pt and the reference electrode was saturated calomel electrode (SCE). The potentials of the Ru/TiO_x as working electrode (1.0-1.7 V vs. RHE) were precisely controlled. ”

On SI page 19-20:

Supplementary Figure 27. Photograph (a) and schematic diagram (b) of the in-situ XPS analysis of Ru/TiO_x .

Supplementary Figure 28. In-situ XPS spectra of Ru/TiO_x during the OER test. In-situ Ru 3d (a), Ti 2p & Ru 3p (b) and O 1s (c) XPS spectra recorded of the as-prepared Ru/TiO_x at applied potential during 1.0-1.7 V vs. RHE.

On SI page 20:

Supplementary Figure 29. Variation of Ru^{n+}/Ru^0 , Ti^{3+}/Ti^{4+} and O_V/O_L (oxygen vacancy/lattice oxygen) ratio from in-situ XPS measurement.

On SI page 39:

Supplementary Table 9. High resolution Ru 3d, Ti 2p and O 1s XPS peak fitting parameters of Ru/TiO_x at applied potential during 1.0-1.7 V vs. RHE.

Sample	Core level	Peak position (eV)	Peak area	FWHM (eV) ^{a)}	
Ru/TiO _x -1.0	Ru 3d _{5/2}	280.12	2593.16	1.07	
		280.77	1836.78	1.17	
	Ru 3d _{3/2}	284.22	1728.77	1.11	
		284.87	1224.52	1.19	
	Ti 2p _{3/2}	457.90	7399.48	1.20	
		458.51	23726.30	1.14	
	Ti 2p _{1/2}	463.90	3699.74	1.21	
		464.51	11863.15	1.54	
	Ru 3p _{3/2}	460.60	1470.22	1.90	
		462.96	2533.76	1.48	
O 1s	529.52	19079.02	1.88		
	530.45	9727.79	1.45		
Ru/TiO _x -1.2	Ru 3d _{5/2}	531.47	8456.91	1.79	
		280.13	2734.80	1.05	
	Ru 3d _{3/2}	280.81	1969.02	1.20	
		284.23	1823.20	1.07	
	Ti 2p _{3/2}	284.91	1312.68	1.21	
		457.90	7001.99	1.23	
	Ti 2p _{1/2}	458.58	22629.02	1.09	
		463.90	3500.99	1.23	
			464.50	11314.51	1.49

Ru/TiO _x -1.4	Ru 3p _{3/2}	460.60	1591.89	1.89
		462.95	2534.37	1.68
	529.55	19282.80	1.86	
	O 1s	530.40	9066.99	1.51
		531.43	8160.46	1.89
	Ru 3d _{5/2}	280.15	2678.28	1.07
		280.83	1945.85	1.44
	Ru 3d _{3/2}	284.25	1785.52	1.08
		284.93	1297.23	1.45
	Ti 2p _{3/2}	457.90	3294.47	1.16
		458.54	15788.83	1.14
	Ti 2p _{1/2}	463.90	1647.24	1.16
		464.54	7894.42	1.54
	Ru/TiO _x -1.6	Ru 3p _{3/2}	460.90	1172.22
463.08			1976.27	1.67
529.60		20864.25	1.72	
O 1s		530.48	9731.28	1.41
		531.49	8407.48	1.98
Ru 3d _{5/2}		280.17	2435.83	1.03
		280.95	1688.01	1.37
Ru 3d _{3/2}		284.27	1557.22	1.05
		285.05	1218.68	1.37
Ti 2p _{3/2}		457.90	5249.07	0.98
		458.60	29718.37	1.10
Ti 2p _{1/2}		463.90	2624.53	0.98
		464.60	14859.18	1.50
Ru/TiO _x -1.7		Ru 3p _{3/2}	460.40	2316.29
	463.15		3039.33	1.52
	529.67	21524.80	1.63	
	O 1s	530.48	9988.22	1.38
		531.52	7457.96	1.89
	Ru 3d _{5/2}	280.21	2506.04	1.04
		281.00	1826.04	1.35
	Ru 3d _{3/2}	284.31	1670.69	1.04
		285.10	1217.36	1.35
	Ti 2p _{3/2}	457.90	5156.78	1.12
		458.63	29558.12	1.07
	Ti 2p _{1/2}	463.90	2578.39	1.12
		464.63	14779.06	1.47
	Ru 3p _{3/2}	460.60	2411.36	1.98
463.27		3225.89	1.89	
526.69		23711.47	1.56	
O 1s		530.46	11822.26	1.27
		531.50	8267.88	2.03

^{a)}FWHM: full-width at the half of the maximum.

Comment #1-6) Ru/TiO_x can achieve high current density of 500 mA cm⁻² under overpotential of 265 mV. Compared with other acidic OER catalysts, what is the main reason for its ability to achieve high current density?

Response: Thanks for the insightful comment. In the original manuscript, we reported that the as-synthesized Ru/TiO_x can reach industrial current densities (500 mA cm⁻²) at low overpotential (265 mV). To illustrate the practical capabilities of Ru/TiO_x in water splitting, a proton exchange membrane water electrolysis (PEMWE) was constructed to evaluate the performance under conditions that are representative of industrial applications (*Supplementary Figs. 21-22, Pages 16-17*). Surprisingly, when using a self-supported Ru/TiO_x electrode as an anode, the PEMWE can operate stable for at least 200 hours under high current density of 500 mA cm⁻², which outperforms other powder-based acidic OER catalysts (*Supplementary Table 9, Page 39*). Combining experimental results and characterization analysis, we revealed the main reason for the enhanced activity and stability of Ru/TiO_x at large current densities as follows: a) **binder-free electrode**: the design of the integrated electrode avoids the coating of binder, allowing the fully exposure the active sites (*Fig. 1 and Supplementary Figs. 1-3*); b) **nanoarray structure**: the design of the nanoarray structure benefits for the electrolyte transport and bubble desorption under high current densities (*Supplementary Figs. 16 and 17*); c) **strong metal-support interactions**: The strong interaction between the metal and the carrier protects the valence state, size and dispersion of Ru, thus ensuring its activity and stability at large current densities (*Supplementary Figs. 26-29; 41-43; Tables 8 and 9*).

Reviewer #2 (Remarks to the Author):

This paper presents an investigation into a method aimed at stabilizing Ru-based catalysts for acidic water oxidation reactions through the use of non-stoichiometric titanium oxides (TiO_x). A significant improvement in the stability and activity of the Ru/ TiO_x catalyst, compared to traditional stoichiometric TiO_2 supports, is reported by the authors. Notably, the synthesis approach is streamlined, involving a single-step process in which Ti foam is employed as a support for the catalyst. This work represents a valuable contribution to the quest for developing cost-effective and stable catalysts for acidic water oxidation. However, the paper falls short in a few key areas, most notably in the depth and rigor of characterization and system-related analyses. Additionally, the computational calculations presented in the study appear overly simplified and, as such, insufficient to robustly support the experimental results. The specific concerns are elaborated upon as follows.

Comment #2-1) Unconvincing Statement on Non-Stoichiometric TiO_2 and Ru Oxidation State:

The current assertions regarding the non-stoichiometric nature of TiO_2 and the oxidation state of Ru under reaction conditions are not compelling. The authors should perform *in situ* X-ray Absorption Spectroscopy (XAS) at the Ti and Ru *K*-edges. This would enable a deeper and more precise understanding of the oxidation states and chemical environments of Ti and Ru under actual reaction conditions, and potentially strengthen the argument presented.

Response: We thank the reviewer very much for the constructive suggestions. It is very important to study the oxidation state changes of Ru and Ti under actual reaction conditions. However, since XAS requires the use of high-energy synchrotron radiation, it is difficult for us to implement *in-situ* XAS testing of the OER catalysts, especially for Ru *K*-edges. Alternatively, the *in-situ* X-ray photoelectron spectroscopy (XPS) has been proved to be an efficient technology to reveal the valence state changes of catalysts during the catalytic process (*Chem. Rev.* 2023, **123**, 6257-6358; *Nat. Energy* 2023, **8**, 372-380; *Nat. Catal.* 2021, **4**, 469-478; *Nat. Commun.* 2023, **14**, 1412; *Nat. Commun.* 2022, **13**, 5448). Therefore, we conducted *in-situ* XPS to study the changes in the oxidation states of Ru and Ti in the catalyst. Through fitting and analysis of *in-situ* XPS data, we compared the peak position changes and the proportion of each valence state of Ru and Ti in detail. As a result, the oxidation state of Ru remains unchanged, while the ratio of $\text{Ti}^{3+}/\text{Ti}^{4+}$ and O_v/O_L (oxygen vacancy/lattice oxygen) slightly changes and remains stable, indicating the critical role of the Ti^{3+} and O_v in stabilizing Ru active sites. The overall conclusion is in good agreement with the *ex-situ* XPS results. The following discussion has been added to the revised manuscript (Page 16 and 24) and Supplementary Information (*Supplementary Figs. 27-29* and *Table 9*, Page 19 and 39) as below:

On page 16: “Afterwards, the chemical environments of Ti and Ru in the Ru/ TiO_x catalyst under actual OER conditions were monitored by *in-situ* XPS (*Supplementary Figs. 27-29*). Significantly, the Ru 3d XPS peaks at 280.1 and 284.2 eV exhibit negligible changes with the applied potential increased from 1.0 to 1.7 V vs. RHE (*Supplementary Fig. 28a* and *Table 9*). Detailed quantitative analysis shows the ratio of $\text{Ru}^{n+}/\text{Ru}^0$ remains almost identical at 0.7 as the voltage increases, which is consistent with the *ex-situ* XPS analysis results (*Supplementary Figs. 26* and *29*). The above results further confirm the stable Ru chemical state in Ru/ TiO_x during the OER process. For the Ti 2p spectra (*Supplementary Fig. 28b* and *Table 9*), the coexistence of Ti^{3+} and Ti^{4+} species was distinguished. Interestingly, as the voltage increases, the ratio of $\text{Ti}^{3+}/\text{Ti}^{4+}$ slightly decreases and remains stable at 0.17, indicating the critical role of the Ti^{3+} in stabilizing Ru active sites. Corresponding to this phenomenon is the O_v/O_L (oxygen vacancy/lattice oxygen) ratio

obtained from the O 1s spectra (Supplementary Figs. 28c and 29). It can be seen that as the voltage increases, the O_V/O_L ratio first decreases and then almost returns to the initial state, manifesting that O_V regeneration is accompanied by the release of oxygen, thereby stabilizing active species³⁹.”

On page 24: “In-situ XPS spectra were measured by ambient pressure XPS end station equipped with a static electrochemical cell at ESCLAB 250Xi (Supplementary Fig. 27). The counter electrode was Pt and the reference electrode was saturated calomel electrode (SCE). The potentials of the Ru/TiO_x as working electrode (1.0-1.7 V vs. RHE) were precisely controlled. ”

On SI page 19-20:

Supplementary Figure 28. In-situ XPS spectra of Ru/TiO_x during the OER. In-situ Ru 3d (a), Ti 2p & Ru 3p (b) and O 1s (c) XPS spectra recorded of the as-prepared Ru/TiO_x at applied potential during 1.0-1.7 V vs. RHE.

On SI page 20:

Supplementary Figure 29. Variation of Ru^{3+}/Ru^0 , Ti^{3+}/Ti^{4+} and O_V/O_L (oxygen vacancy/lattice oxygen) ratio from in-situ XPS measurement.

On SI page 39:

Supplementary Table 9. High resolution Ru 3d, Ti 2p and O 1s XPS peak fitting parameters of Ru/TiO_x at applied potential during 1.0-1.7 V vs. RHE.

Sample	Core level	Peak position (eV)	Peak area	FWHM (eV) ^{a)}
Ru/TiO _x -1.0	Ru 3d _{5/2}	280.12	2593.16	1.07
		280.77	1836.78	1.17
		284.22	1728.77	1.11
	Ru 3d _{3/2}	284.87	1224.52	1.19
		457.90	7399.48	1.20
	Ti 2p _{3/2}	458.51	23726.30	1.14
		463.90	3699.74	1.21
	Ti 2p _{1/2}	464.51	11863.15	1.54
		460.60	1470.22	1.90
	Ru 3p _{3/2}	462.96	2533.76	1.48
		529.52	19079.02	1.88
	O 1s	530.45	9727.79	1.45
		531.47	8456.91	1.79
Ru/TiO _x -1.2	Ru 3d _{5/2}	280.13	2734.80	1.05
		280.81	1969.02	1.20
		284.23	1823.20	1.07
	Ru 3d _{3/2}	284.91	1312.68	1.21
		457.90	7001.99	1.23
	Ti 2p _{3/2}	458.58	22629.02	1.09
		463.90	3500.99	1.23
	Ti 2p _{1/2}	464.50	11314.51	1.49
		460.60	1591.89	1.89
	Ru 3p _{3/2}	462.95	2534.37	1.68
		529.55	19282.80	1.86
	O 1s	530.40	9066.99	1.51
		531.43	8160.46	1.89
Ru/TiO _x -1.4	Ru 3d _{5/2}	280.15	2678.28	1.07
		280.83	1945.85	1.44
		284.25	1785.52	1.08
	Ru 3d _{3/2}	284.93	1297.23	1.45
		457.90	3294.47	1.16
	Ti 2p _{3/2}	458.54	15788.83	1.14
		463.90	1647.24	1.16
	Ti 2p _{1/2}	464.54	7894.42	1.54
		460.90	1172.22	1.98
	Ru 3p _{3/2}	463.08	1976.27	1.67
		529.60	20864.25	1.72
	O 1s	530.48	9731.28	1.41
		531.49	8407.48	1.98
Ru/TiO _x -1.6	Ru 3d _{5/2}	280.17	2435.83	1.03
		280.95	1688.01	1.37

Ru/TiO _x -1.7	Ru 3d _{3/2}	284.27	1557.22	1.05
		285.05	1218.68	1.37
	Ti 2p _{3/2}	457.90	5249.07	0.98
		458.60	29718.37	1.10
	Ti 2p _{1/2}	463.90	2624.53	0.98
		464.60	14859.18	1.50
	Ru 3p _{3/2}	460.40	2316.29	1.92
		463.15	3039.33	1.52
	O 1s	529.67	21524.80	1.63
		530.48	9988.22	1.38
	Ru 3d _{5/2}	531.52	7457.96	1.89
		280.21	2506.04	1.04
	Ru 3d _{3/2}	281.00	1826.04	1.35
		284.31	1670.69	1.04
	Ti 2p _{3/2}	285.10	1217.36	1.35
		457.90	5156.78	1.12
	Ti 2p _{1/2}	458.63	29558.12	1.07
		463.90	2578.39	1.12
	Ru 3p _{3/2}	464.63	14779.06	1.47
		460.60	2411.36	1.98
O 1s	463.27	3225.89	1.89	
	526.69	23711.47	1.56	
O 1s	530.46	11822.26	1.27	
	531.50	8267.88	2.03	

^{a)}FWHM: full-width at the half of the maximum.

Comment #2-2) Necessity for Real-World Testing Conditions:

The long stability tests presented in this work are insufficiently related to practical, industrial scenarios. The authors should conduct stability tests under conditions that are representative of industrial applications, specifically within a real membrane electrode assembly (MEA) that includes a proton exchange membrane, and at industrially relevant current densities. The use of 10 mA/cm² for testing seems to fall short of the conditions that this material would face in a real-world application.

Response: We appreciate the important comment of the reviewer very much. Accordingly, the stability of as-prepared Ru/TiO_x in PEM electrolyzers was further evaluated. Specifically, we constructed a PEMWE electrolyser using Ru/TiO_x as the anode catalyst for OER, commercial Pt/C as the cathode catalyst for HER and a proton exchange membrane (Nafion 212). As a result, the PEMWE with Ru/TiO_x as an anode shows 1.71 V at 1.0 A cm⁻², maintains its voltage during 200 h operation at 500 mA cm⁻². Such performance significantly outperforms those with RuO₂ anode, and is comparable to the state-of-the-art catalysts, which demonstrates that Ru/TiO_x is potentially to be utilized in industrial applications as an acidic OER catalyst in PEMWE. The following discussion has been added to the revised manuscript (Page 13 and 25) and Supplementary Information (*Supplementary Figs. 21-22 and Table 6*, Page 16-17 and 36) as below:

On page 13: “To illustrate the practical capabilities of Ru/TiO_x in water splitting, a PEMWE was constructed to evaluate the performance under conditions that are representative of industrial applications

(Supplementary Fig.21 and 22). As shown in Supplementary Fig. 22a, the PEMWE with Ru/TiO_x as an anode shows 1.71 V at 1.0 A cm⁻², which is 0.23 V lower compared with RuO₂ anode. Besides, the PEMWE with Ru/TiO_x as an anode well maintains its voltage ($\Delta E < 0.01$ V) during 200 h operation at 500 mA cm⁻², comparable to the recently reported catalysts (Supplementary Fig. 22b and Table 6). In addition, the stability significantly outperforms those with RuO₂ anode, which shows $\Delta E > 0.2$ V decay within 100 h (Supplementary Fig. 22b), further highlights the superiority of self-supported Ru/TiO_x in OER at high current densities.”

On page 25: “For electrolyser tests, a self-made cell was used as the PEMWE device and a cation exchange membrane (Nafion 212) as the membrane electrolyte. The membrane electrode assembly (MEA) was prepared by pressing the cathodes (20% Pt/C sprayed on the Nafions 212 membrane) and anodes (self-supported Ru/TiO_x). During the test, the cell was maintained at 60°C, and the pre-heated deionized water was fed to the anode by a peristaltic pump. All the data of PEMWE were not iR corrected and displayed as raw data.”

On SI pages 16-17:

Supplementary Figure 21. Optical photo (a) and schematic diagram (b) of the proton-exchange membrane water electrolyzers (PEMWE).

Supplementary Figure 22. a, Polarization curves of PEMWE utilizing the as-synthesized Ru/TiO_x or commercial RuO₂ as an anode and commercial Pt/C as a cathode. b, The corresponding stability test of the PEMWE at 500 mA cm⁻².

On SI page 36:

Supplementary Table 2. Comparison of the PEM electrolyzer performance with those previously reported.

Anode catalysts	Cell voltage (V)	Stability	Reference
Ru/TiO _x	1.71 V @ 1 A cm ⁻²	0.5 A cm ⁻² for 200 h	This work
RuO ₂	1.94 V @ 1 A cm ⁻²	0.5 A cm ⁻² for < 50 h	This work
Nb _{0.1} Ru _{0.9} O ₂	1.69 V @ 1 A cm ⁻²	0.3 A cm ⁻² for 100 h	Joule 7 , 558-573 (2023)
Y ₂ MnRuO ₇	1.51 V @ 0.2 A cm ⁻²	0.2 A cm ⁻² for 24 h	Nat. Commun. 14 , 2010 (2023)
Nd _{0.1} RuO _x	1.595 V @ 0.05 A cm ⁻²	0.05 A cm ⁻² for 50 h	Adv. Funct. Mater. 33 , 2213304 (2023)
IrO _x /Zr ₂ ON ₂	1.927 V at 2.0 A cm ⁻²	1.0 A cm ⁻² for 50 h	Adv. Funct. Mater. 33 , 2301557 (2023)
RuO ₂ /Defect-TiO ₂	1.74 V @ 1.5 A cm ⁻²	1.0 A cm ⁻² for 6 h	ACS Catal. 12 , 9437-9445 (2022)
Strained-RuO ₂ /ATO	1.51 V @ 1 A cm ⁻²	0.5 A cm ⁻² for 40 h	Adv. Sci. 9 , 2201654 (2022)
W _{0.2} Er _{0.1} Ru _{0.7} O _{2-δ}	-	0.1 A cm ⁻² for 120 h	Nat. Commun. 11 , 5368 (2020)

Comment #2-3) Need for Pourbaix Diagram in Computational Study:

The computational chemistry aspects of the work could be enhanced significantly. The authors should utilize the Pourbaix diagram in their computational study. This would help to understand the thermodynamic stability of the various possible oxidation states of Ru and Ti under different pH and potential conditions, and could elucidate the origin of the observed high stability of the material.

Response: We appreciate for the important suggestions. Accordingly, we calculated and obtained the Pourbaix diagram to study the thermodynamic stability of the various possible oxidation states of Ru and Ti under different pH and potential conditions. From the calculated Pourbaix diagram, it can be seen that the stable oxidation states of Ti and Ru are TiO₂ and RuO₂ under the experimental potential condition, respectively. These results provide the basis for our theoretical model. The following description has been added to the revised manuscript (Page 19) and Supplementary Information (*Supplementary Fig. 36*, Page 26) as below:

On page 19: “The Pourbaix diagrams were constructed to study the oxidation states of Ru and Ti under different pH and potential conditions (*Supplementary Fig. 36*). From the calculated Pourbaix diagrams, it can be seen that the stable oxidation states of Ti and Ru are TiO₂ and RuO₂ under the experimental potential condition, respectively.”

On SI page 26:

Supplementary Figure 36. Calculated Pourbaix diagrams of Ti (a) and Ru (b) systems.

Comment #2-4) Simplistic Model of Non-Stoichiometric TiO₂:

The non-stoichiometric TiO₂ model with a single oxygen vacancy, as used in this study, appears overly simplistic. The authors should consider adopting a more complex and realistic model of non-stoichiometric TiO₂. Optimization using molecular dynamics simulations may offer a more accurate representation of the actual material and its behavior under operational conditions.

Response: Thanks for the important comment. Based on the reviewers' suggestions, we reanalyzed the 'Mechanism analysis of OER activity' section (Pages 18-21). The main changes include: 1) The *ab initio* molecular dynamic (AIMD) was carried out to obtain stable structures (Figs. 5a-b and Supplementary Figs.36-40). The AIMD simulations at 300 K for 10 ps with a time step 1fs; 2) On this basis, Bader charge, PDOS, TDOS and adsorption energies of OER intermediates were analyzed (Figs. 5c-e Supplementary Figs.41-47); 3) Based on the conclusions of *in-situ* XPS (oxygen vacancies can be regenerated during the OER process and their concentration will be stable), we further constructed a structure with two oxygen vacancies in TiO_x (Ru/TiO_x) to study its charge distribution and OER mechanism. We believe that the above improvements can offer a more accurate representation of the actual material and its behavior under operational conditions. The main revisions are shown as below:

On page 19: "To obtain stable Ru/TiO_x structures, we carried out *ab initio* molecular dynamic (AIMD) (Supplementary Figs. 37-40). The AIMD simulations at 300 K for 10 ps with a time step 1 fs. These results provide the basis for our theoretical models."

On page 19: "It has been reported that catalysts tend to pre-adsorb oxygen species and generate amorphous nonstoichiometric oxide layers under the OER working condition^{13,41}. Therefore, the Ru₅ cluster will also be present as oxides during OER. Since the existence of oxygen vacancies will affect the oxidation state of Ru, we constructed different models (V₁₀-RuO/TiO_x, V₂₀-RuO/TiO_x, P-RuO/TiO₂ and P-RuO_v/TiO₂, V₂₀ denotes two V_O) to study the OER process (Supplementary Fig. 40). Whether it is the initial Ru₅ cluster or RuO_x after pre-oxidation, the introduction of V_O helps to maintain the Ru atom at the interface in a lower valence state. Bader charge analysis shows that the charge of the Ru atom at the interface increases from 7.72 e⁻ in P-Ru₅/TiO₂ to 7.87 e⁻ and 8.02 e⁻ in V₁₀-Ru₅/TiO_x and V₂₀-Ru₅/TiO_x, respectively. The results obtained for the pre-oxidized structures are in good agreement with this conclusion (Supplementary Fig.

41). The electron distribution are further proved by the charge density differences (Supplementary Fig. 42), the significant charge redistributions in V_{10} -RuO/TiO_x indicate stronger interaction between them than P-RuO_{1.6}/TiO_x.”

On page 37:

Fig. 5 | Mechanism analysis of Ru/TiO_x towards acidic OER. **a**, Atomic structure of Ru₅ cluster adsorbed on the TiO₂ (110) surface with single oxygen vacancy (V_{10} -Ru₅/TiO_x). **b**, The adsorption energy of oxygen (ΔE^*_{O}) on the Ru sites and the corresponding Ru-Ru bond length value (d_{Ru-Ru}). The insets are the atomic structures of $*O$ adsorbed on the structures of Ru₅ cluster adsorbed on the pristine TiO₂ (P-Ru₅/TiO₂) and V_{10} -Ru₅/TiO_x. **c**, Gibbs free energy profile of OER for P-RuO_{1.6}/TiO₂, V_{10} -RuO/TiO_x and V_{20} -RuO/TiO_x (V_{20} denotes 2 V_{Os}). The insets are the atomic structures of V_{10} -RuO/TiO_x and V_{20} -RuO/TiO_x. **d**, Projected density of states (PDOS) and band center of Ru d-state for P-RuO_{1.6}/TiO₂ and V_{10} -RuO/TiO_x. **e**, Crystal Orbital Hamilton Population (COHP) for the adsorbed $*O$ of Ru-O in P-RuO_{1.6}/TiO₂ and V_{10} -RuO/TiO_x.

On SI page 26:

Supplementary Figure 37. The total energy of $V_{10}\text{-TiO}_x$ (a), $P\text{-Ru/TiO}_2$ (b), $V_{10}\text{-Ru/TiO}_x$ (c) and $V_{20}\text{-Ru/TiO}_x$ (d) as a function of molecular dynamic (MD) time at a temperature of 300 K. (V_{10} and V_{20} denotes 1 and 2 oxygen vacancies, respectively; P denotes perfect structure without oxygen vacancy)

On SI page 28:

Supplementary Figure 40. Theoretical calculation models of $\text{RuO}_x/\text{TiO}_x$. Top view and side view of $P\text{-RuO/TiO}_2$ (a,b), $P\text{-RuO}_{1.6}/\text{TiO}_2$ (c,d), $V_{10}\text{-RuO/TiO}_x$ (e,f) and $V_{20}\text{-RuO/TiO}_x$ (g,h), respectively. (The gray, blue and red balls represent Ru, Ti and O atoms, respectively).

Comment #2-5) Comparison of Experimental and Theoretical Turnover Frequency (TOF):

While the geometry current density improvement of Ru/TiO_x is noteworthy, the ECSA (Electrochemical Surface Area) normalized improvement does not appear to be significant. Given that reaction energies have been calculated in the study, the authors should also calculate the theoretical turnover frequency (TOF) and

compare it with the experimental TOF. This would give a more comprehensive understanding of the catalytic efficiency of the material system under investigation.

Response: We thank the reviewer for the constructive comment. Turnover frequency (TOF) is the most reasonable description of intrinsic activity, but mostly difficult to assess. Experimentally, we have calculated the TOF values based on the assumption that all Ru metal atoms are active sites, and the number of metal atoms are obtained from ICP-MS results (Equations (1)-(3)). Calculating TOF through this method is the most widely used, and the resulting TOF value is considered as ‘Bulk TOF’ (*Science* 2023, **380**, 609-616; *Nat. Catal.* 2021, **4**, 1012-1023; *Nat. Commun.* 2021, **12**, 4587; *Adv. Mater.* 2020, **32**, 2002235). As a result, the bulk TOF of Ru/TiO_x (1.96 s⁻¹) at 1.53 V vs. RHE is the highest when compared with annealed RuO_x/TiO₂ (0.82 s⁻¹), commercial RuO₂ (0.32 s⁻¹) and other representative OER catalysts tested in various acidic media (*Supplementary Fig. 14 and Table 4*). Since the reviewer mentioned that ‘the ECSA (Electrochemical Surface Area) normalized improvement does not appear to be significant’, we further provided a bar graph of ECSA-normalized current densities and bulk TOF values at an overpotential of 300 mV, which shows similar intrinsic activity trend (*Supplementary Fig. 14b*).

$$\text{TOF} = \frac{\# \text{ Total Oxygen Turn Overs per geometric area}}{\# \text{ Active sites per geometric/real area}} \quad (1)$$

Total oxygen turn overs per geometric area

$$\begin{aligned} &= (j \frac{\text{mA}}{\text{cm}^2}) (\frac{1 \text{C s}^{-1}}{1000 \text{ mA}}) (\frac{1 \text{ mol O}_2}{4 \text{ mol e}^-}) (\frac{6.022 \times 10^{23} \text{ O}_2 \text{ atoms}}{1 \text{ mol O}_2}) (\frac{1 \text{ mol e}^-}{96485.3 \text{ C}}) \\ &= 1.56 \times 10^{15} \left(\frac{\text{O}_2 \text{ s}^{-1}}{\text{cm}^2} \right) \text{ per } \left(\frac{\text{mA}}{\text{cm}^2} \right) \end{aligned} \quad (2)$$

Active sites per geometric area

$$= \left(\frac{\text{mass loading of Ru } (\text{g}/\text{cm}^2)}{\text{Ru Mw } (\text{g}/\text{mol})} \right) \left(\frac{6.022 \times 10^{23} \text{ Ru atoms}}{1 \text{ mol Ru}} \right) \quad (3)$$

Another way to calculate the TOF value is based on the assumption that the number of active sites is inferred from the electrochemically active surface area (ECSA). The active sites per real surface area is calculated through unit cell volume and the number of active atoms (Equations (1) and (4)) (*Science* 2023, **380**, 609-616; *Angew. Chem. Int. Ed.* 2014, **53**, 14433-14437). Therefore, we also calculated the TOF value using the unit cell parameters obtained by DFT modeling and the ECSA value calculated by C_{dl} (*Supplementary Fig. 11*). The TOF of Ru/TiO_x is calculated to be 2.192 s⁻¹ at an overpotential of 300 mV based on ECSA, which is the highest among the as-synthesized samples (*Supplementary Table 5*). The difference in TOF of different catalysts calculated by ECSA becomes smaller, which is due to the higher electrochemical surface area of Ru/TiO_x resulting from the nanoarray structure. Since this method involves the calculation of theoretical active sites through unit cell, we believe that the ECSA TOF value can represent the theoretical TOF value to a certain extent.

Active sites per real surface area:

$$= \left(\frac{\text{number of active sites / unit cell}}{\text{unit cell volume}} \right)^{2/3} \quad (4)$$

We learned that a computationally derived TOF of catalytic cycles can be calculated through transition state

theory (*ACS Catal.* 2022, **12**, 9058-9073). However, because our models are very complex and involve Ru clusters, the calculation is very difficult to be done. Alternatively, we have compared the TOF values obtained by different methods in detail and compared them with literature values, with the purpose of revealing and comparing the intrinsic activities. The corresponding discussion of TOF has been added to the revised manuscript (Page 10) and the Supplementary Information (*Supplementary Fig. 14 and Tables 4-5*, Page 5, 12 and 35) as below:

On page 10: “The intrinsic activities of Ru/TiO_x were further assessed based on turnover frequencies (TOFs) at different overpotentials, which are among the highest when compared with representative OER catalysts in various acidic media (*Supplementary Fig. 14a and Table 4*). For example, the TOF of Ru/TiO_x is calculated to be 1.960 s⁻¹ at an overpotential of 300 mV based on the total loading mass, which increases to 2.192 s⁻¹ when calculated based on ECSA (*Supplementary Table 5*). We also provided a bar graph of ECSA-normalized current densities and TOF values at an overpotential of 300 mV, which shows similar activity trend (*Supplementary Fig. 14b*). The above results prove that Ru/TiO_x shows the highest intrinsic activity per site.”

On SI page 5: “**Calculation of turnover frequency (TOF):** The TOF value based on inductively coupled plasma mass spectrometry (ICP-MS) results (bulk TOF) and electrochemical active surface area (ECSA) values (ECSA TOF) were calculated and compared. The Bulk TOF value was calculated by following formula:

$$\text{TOF} = \frac{\# \text{ Total Oxygen Turn Overs per geometric area}}{\# \text{ Active sites per geometric area}} \quad (1)$$

Total oxygen turn overs per geometric area

$$\begin{aligned} &= \left(j \frac{\text{mA}}{\text{cm}^2} \right) \left(\frac{1 \text{C s}^{-1}}{1000 \text{ mA}} \right) \left(\frac{1 \text{ mol O}_2}{4 \text{ mol e}^-} \right) \left(\frac{6.022 \times 10^{23} \text{ O}_2 \text{ atoms}}{1 \text{ mol O}_2} \right) \left(\frac{1 \text{ mol e}^-}{96485.3 \text{ C}} \right) \\ &= 1.56 \times 10^{15} \left(\frac{\text{O}_2 \text{ s}^{-1}}{\text{cm}^2} \right) \text{ per } \left(\frac{\text{mA}}{\text{cm}^2} \right) \end{aligned} \quad (2)$$

Active sites per geometric area

$$= \left(\frac{\text{mass loading of Ru } \left(\frac{\text{g}}{\text{cm}^2} \right)}{\text{Ru Mw } \left(\frac{\text{g}}{\text{mol}} \right)} \right) \left(\frac{6.022 \times 10^{23} \text{ Ru atoms}}{1 \text{ mol Ru}} \right) \quad (3)$$

The ECSA TOF value was calculated by Equation (2) and the following formula:

$$\text{TOF} = \frac{\# \text{ Total Oxygen Turn Overs per geometric area}}{\# \text{ Active sites per real area}} \quad (4)$$

Active sites per real surface area:

$$= \left(\frac{\text{number of active sites / unit cell}}{\text{unit cell volume}} \right)^{2/3} \quad (5) \quad ”$$

On SI page 12:

Supplementary Figure 14. a, Potential-dependent turnover frequency (TOF) curves. **b**, The corresponding bar graph of the ECSA-normalized current density (blue) and TOF values (red) at the overpotential of 300 mV.

On SI page 35:

Supplementary Table 4. TOF of Ru/TiO_x with previously reported OER catalysts in acid.

Catalysts	Electrolyte	Overpotential (mV)	TOF (s ⁻¹)	Reference
Ru/TiO _x	0.5 M H ₂ SO ₄	270 300	1.707 1.960	This work
Annealed RuO _x /TiO ₂	0.5 M H ₂ SO ₄	300	0.820	This work
Com. RuO ₂	0.5 M H ₂ SO ₄	300	0.322	This work
SnRuO _x	0.5 M H ₂ SO ₄	250	0.63	Nat. Commun. 14 , 843 (2023)
Rh-RuO ₂ /Graphene	0.5 M H ₂ SO ₄	300	1.74	Nat. Commun. 14 , 1412 (2023)
high-loading Ir single atoms with d -band holes	0.1 M HClO ₄	216	0.599	Angew 135 , 202308082 (2023)
Ru ₅ W ₁ O _x	0.5 M H ₂ SO ₄	300	0.163	Nat. Commun. 13 , 4871 (2022)
Cr-SrIrO ₃	0.1 M HClO ₄	300	0.208	Nano Energy 102 , 107680 (2022)
Ru/MnO ₂	0.1 M HClO ₄	165	0.331	Nat. Catal. 4 , 1012-1023 (2021)
Ru ₁ Ir ₁ O _x	0.5 M H ₂ SO ₄	300	0.47	Adv. Energy Mater. 11 , 2102883 (2021)

Supplementary Table 5. TOF of catalysts using different normalization methods.

Catalysts	Overpotential (mV)	Bulk TOF (s ⁻¹)	ECSA TOF (s ⁻¹)
Ru/TiO _x	270	1.707	1.835
	300	1.960	2.192
Annealed RuO _x /TiO ₂	300	0.820	1.640
Com. RuO ₂	300	0.322	1.520

Comment #2-6) Discrepancy Between Electrical Conductivity and DFT Calculations:

The authors mention that the "substoichiometric phase Ti₂O₃ exhibits high electrical conductivity". However, their DFT calculations suggest the TiO_x material is a semiconductor with a large band gap. The authors should resolve this apparent contradiction. A comprehensive explanation is necessary to clarify this

discrepancy and align the DFT calculations with the claimed electrical properties of the material.

Response: Thank you very much for the important comments. In the previous version of our manuscript, we only show the projected density of states (PDOS) of Ru *d*-orbital. There is a large band gap. However, for the total density of state (TDOS) of RuO/TiO_x systems, there is no band gap (*Supplementary Fig. 47*). More importantly, there are more states around the Fermi level in the V-RuO/TiO_x systems than P-RuO/TiO_x, indicating a high electrical conductivity. It is possible that oxygen vacancy induce a stronger interaction at the interface and changes the electronic structure of the RuO/TiO_x system. The analysis of TDOS for RuO/TiO_x systems have been added in revised manuscript (Page 20) and Supplementary Information (*Supplementary Fig. 47*, Page 31).

On page 20: “The spin-polarized total density of states (TDOS) and projected density of states (PDOS) of Ru’s *d*-orbitals for the P-RuO_{1.6}/TiO₂ and the V-RuO/TiO_x models are shown in Fig. 5d and Supplementary Figs. 46-47). The analysis of the *d*-band center results of Ru atoms at the interface shows a shift from -1.01 eV (P-RuO_{1.6}/TiO₂) to -1.33 eV (V₁₀-RuO/TiO_x) and -1.66 eV (V₂₀-RuO/TiO_x) upon the formation of V_O (Fig. 5d and Supplementary Fig. 46), indicating the weaker binding to adsorbates according to the classical *d*-band theory⁴⁴. More importantly, the V-RuO/TiO_x systems have more states around fermi level than P-RuO/TiO_x, indicating high electrical conductivity (*Supplementary Fig. 47*). It is possible that V_O induce a stronger interaction at the interface and changes the electronic structure of the RuO/TiO_x system.”

On SI page 31:

Supplementary Figure 47. Total density of states (TDOS) for P-RuO_{1.6}/TiO₂ (a), V₁₀-RuO/TiO_x (b) and V₂₀-RuO/TiO_x (c).

Reviewer #3 (Remarks to the Author):

The authors report a Ti oxides (TiO_x) supported Ru electrocatalyst (Ru/TiO_x) working in an acidic electrolyte. The Ru/TiO_x exhibits low OER overpotentials of 174, 209, and 265 mV to reach current densities of 10, 100, and 500 mA cm^{-2} , respectively, and is stable for 900 h under 10 mA cm^{-2} in 0.5 M H_2SO_4 . Their theoretical calculations indicate that oxygen vacancy (V_O) defects at the Ru-TiO_x interface induced a charge accumulation at Ru sites, thereby preventing the aggregation and over-oxidation of Ru during the OER process. It is, generally, an interesting work. However, the overall novelty of the work is not sufficient for publication in *Nature Communications*. In fact, many reports have already shown it an efficient way to improve the acidic OER stabilities of Ru-based materials by introducing electron-donated supports to prevent the over-oxidation of Ru sites during the reaction (*Nature Catalysis* 2019, **2**, 304-313; *Angew. Chem. Int. Ed.* 2022, **61**, e202202519). The effectiveness of this strategy has also been confirmed in V_O defective TiO_2 supported RuO_2 catalyst (*ACS Catal.* 2022, **12**, 9437-9445). The originality of the work is somehow limited as previous Ru/TiO_2 has been made. Moreover, from catalytic performance perspective, the activity and stability of Ru/TiO_x for acidic OER are not particularly better than those recently reported Ru-based catalysts (*Nature Commun.* 2023, **14**, 843/1412/2517; *Nat. Mater.* 2023, **22**, 100-108). Some other comments for authors to consider:

Response: Thank you very much for your important comments. We have carefully read the literature recommended by the reviewers and other recent advances in the field of acidic OER, and revised the manuscript based on the suggestions of all reviewers. On this basis, we believe that the one-step method to prepare Ru/TiO_x stands out for *catalyst design principles*, *synthesis strategies* and *mechanism investigation*. In short, we believe that both *the one-step strategy to synthesize integrated electrode with ultrahigh OER performance under industrial current densities* and *the in situ introduction of oxygen vacancy to induce charge redistribution*, to the best of our knowledge, are significant breakthroughs in the field

Our main improvements to the manuscript: a) **Performance evaluation under industrial conditions:** a proton exchange membrane water electrolysis (PEMWE) was constructed to evaluate the performance under conditions that are representative of industrial applications. The PEMWE with Ru/TiO_x as the anode can operate under 500 mA cm^{-2} for > 200 h, which is superior to those of most recently developed OER catalysts (*Supplementary Figs. 21-22 and Table 6*); b) **Chemical states-activity relationship establishment:** We further provide the precise understanding of the oxidation states under actual reaction conditions by in-situ X-ray photoelectron spectroscopy (XPS). Through in-situ XPS analysis, we further revealed the role of Ti^{3+} and oxygen vacancies in stabilizing Ru site during the OER process. (*Supplementary Figs. 27-29 and Table 9*); c) **Deep understanding of the OER mechanism:** In the density functional theory (DFT) calculations part, the Pourbaix diagrams were constructed, and the structures of Ru/TiO_x were optimized using *ab initio* molecular dynamics (MD) simulations, which may offer a more accurate representation of the actual catalyst and its behavior under operational conditions (*Supplementary Figs. 36-47*).

Our response to the Reviewer's concern on the novelty of the Ru/TiO_x : To the best of our knowledge, this is indeed **the first report** of using one-step method to synthesize integrated electrode with ultrahigh OER performance under large current densities, achieving industrial standard. More importantly, unlike most methods that optimize OER performance by introducing additional oxygen vacancies, our proposed strategy can *in situ* introduce oxygen vacancies during the material growth process, which is also **reported**

for the first time. We note that the first and second reviewer also recognize our preparing strategy by writing that “*The strategy proposed by the author is very interesting, and the catalysts have been extensively investigated both experimentally and theoretically*” and “*Notably, the synthesis approach is streamlined*”(please see more details in the first and second reviewer’s comments). The concepts of *in situ* metal-support interaction and fabrication of nonstoichiometric compounds may not only offer paths to the next-generation OER catalysts, but also illustrate a promising way to design active sites for nanocatalysts across the wide range of conceivable systems.

Comment #3-1) The potential of reference electrode, Ag/AgCl, needs to be calibrated by a reversible hydrogen electrode (RHE) at first, according to which the reported OER potentials on the RHE scale will be more credible. As well-known, the pH value of 0.5 M H₂SO₄ is around 0.3 rather than 0.0. In the method section, authors do not report the pH value of electrolyte used for the potential conversion based on the equation (Line 511, page 24).

Response: We appreciate the reviewer’s constructive comment very much. We fully agree that measuring and reporting the pH value of the electrolyte is necessary for potentiometric calibration. In fact, we used a pH meter (Mettler Toledo, Germany) to accurately measure the pH value of the 0.5 M H₂SO₄ electrolyte solution before performing all electrochemical tests. We have added the pH reporting in details on Page 24 of the revised manuscript as below:

On page 24: “*Electrochemical measurements were performed with a workstation in a typical three-electrode configuration consisting of a Pt plate (the counter electrode), an Ag/AgCl electrode (the reference electrode) and the active material (the working electrode) in 0.5 M H₂SO₄ solution. The pH of the electrolyte, 0.30 ± 0.01, was measured using a pH-meter (Mettler Toledo, Germany) before each electrochemical test.*”

Comment #3-2) Also for the reported potentials, are the values already *iR* corrected? Please, check it and if they are *iR* corrected change the axis to *E-iR* and report details for *iR* correction in the method section and how much is corrected. The results change a lot if the graphs are corrected or not. Please provide non-*iR* corrected data for Fig. 2a in the SI.

Response: All the electrochemical test data (including the LSV and chronoamperometric curves) we presented in the original manuscript were without *iR* compensation. To eliminate electrolyte resistance, we conducted 95% *iR* correction on all LSV curves and compared the overpotential before and after *iR* correction. The following description has been added to the revised manuscript (Page 8 and 24) and Supplementary Information (*Supplementary Fig. 9*, Page 10) as below:

On page 8: “*As can be seen from the *iR*-free linear sweep voltammetry (LSV) curves, Fig. 2a, the Ru/TiO_x exhibits the highest performance of all samples, requiring overpotentials of only 174, 209, and 226 mV to achieve current densities of 10, 100, and even 200 mA cm⁻²_{geo}, respectively, significantly outperforming the other control samples, including the state-of-the-art commercial RuO₂/TiO₂ and IrO₂/TiO₂. To eliminate electrolyte resistance, the LSV curves with 95 % *iR*-compensation were also shown in Fig. 2a. The measured overpotentials before and after *iR* correction are summarized in *Supplementary Fig. 9*. ”*

On page 24: “*The overpotential values reported in the manuscript are all obtained through the LSV curves without *iR* correction. For comparison, the LSV curves with 95 % *iR* (*i*, current; *R*, resistance)*

compensation were also reported.”

On page 34:

Fig. 2 | Electrochemical oxygen evolution reaction (OER) activity in 0.5 M H₂SO₄ solutions. *a*, OER polarization curves (both *iR*-corrected and *iR*-free), *b*, Tafel plots derived from the LSV curves (solid line) and the steady-state polarization curves (scatters) (values in parentheses were derived from steady-state polarization curves) and *c*, electrochemical impedance spectroscopy (EIS) of Ru/TiO_x, annealed RuO_x/TiO₂, commercial RuO₂/TiO₂, commercial IrO₂/TiO₂ and TiO₂. (Com. RuO₂/TiO₂ and com. IrO₂/TiO₂ denotes commercial RuO₂/TiO₂ and commercial IrO₂/TiO₂, respectively). *d*, Comparison of overpotentials *without iR correction* at 10 and 100 mA cm⁻² for Ru/TiO_x, annealed RuO_x/TiO₂, com. RuO₂/TiO₂ and com. IrO₂/TiO₂. *e*, Comparison of the overpotentials of Ru/TiO_x and state-of-the-art Ru/Ir-based electrocatalysts at 10 mA cm⁻² in acidic media.

On SI page 10:

Supplementary Figure 9. Overpotentials of different samples to reach 10 mA cm⁻² (a) and 100 mA cm⁻² (b) before (smooth) and after (grid filled) *iR* correction.

Comment #3-3) The observed low overpotential may also be due to the partial oxidation of water to hydrogen peroxide in addition to the O₂ evolution (*Sci Bull*, 2023, 68, 613-621; *J. Phys. Chem. Lett.* 2015, 6, 4224-4228). The authors need to prove more convincing data to support their claims.

Response: We thank the reviewer for the constructive comment. We have carefully read the literatures recommended by the reviewer (*Sci Bull*, 2023, 68, 613-621; *J. Phys. Chem. Lett.* 2015, 6, 4224-4228) as well as the literatures on the H₂O₂ production by electrochemical water oxidation (*Adv. Energy Mater.* 2022, 12, 2201466; *Chem* 2021, 7, 38-63; *Nat. Commun.* 2022, 13, 7256). We will try to address this comment from both theoretical and experimental perspectives as below:

Theoretically: the water oxidation reaction (WOR) can proceed through one-electron, two-electron, and four-electron oxidative steps (**Fig. R1**)¹. One-electron WOR produces hydroxyl radical (•OH) and two-electron WOR generates H₂O₂ directly (Equations (1)-(2)). Four-electron WOR, namely OER, produces O₂ (Equations (3)-(6)). In view of thermodynamics, OER is most favorable among the three different WOR pathways due to the lowest equilibrium potential ($E_0 = 1.23$ V vs. RHE). The production of H₂O₂ via H₂O dissociation requires a higher potential ($E_0 = 1.76$ V vs. RHE) than the production of O₂, which makes it more difficult to produce H₂O₂ selectively¹⁻³. Moreover, the generated H₂O₂ can be further oxidized to O₂ under the positive potential. In our work, the voltage range we apply to the catalyst (1.1-1.6 V vs. RHE) has not yet reached the theoretical voltage for generating H₂O₂ (1.76 V vs. RHE). So theoretically, only O₂ can be produced during the electrochemical test of the catalyst, and H₂O₂ will not be produced.

Fig. R1. Schematic diagrams of three different reaction pathways for the water oxidation on the catalytic surface¹.

The two-electron WOR (2e-WOR):

The four-electron WOR (4e-WOR):

Experimentally: For the electrochemical H₂O₂ production through 2e-WOR, there are mainly three methods to quantify the produced H₂O₂: ultraviolet-visible (UV-Vis) spectrophotometry, titration, and colorimetric strips due to their convenience and relatively high quantification accuracy within suitable concentration ranges^{1,2,4}. Here, we adopt the titanium sulfate spectrophotometric method used in the literature recommended by the reviewer to quantify the concentration of H₂O₂⁴. In detail, the exact concentration of the generated H₂O₂ was determined by the titration process of titanium sulfate based on the following reaction⁴⁻⁶:

A reagent solution was prepared by adding 11.52 g of titanium sulfate and 25.55 g of concentrated sulfuric acid to 100 ml of distilled water. Dilute the 30% H₂O₂ solution to form different concentration of H₂O₂ of 4.48×10^{-3} , 2.80×10^{-3} , 5.60×10^{-4} , 1.10×10^{-4} , 5.60×10^{-5} M. Take 9.0 ml of the standard solution and dilute it to 10.0 ml with the reagent solution to obtain a series of orange solutions (**Fig. R2a**). The concentration of peroxide titanium complex, which has an absorbance peak at 410 nm in contrast to the colorless Ti⁴⁺, was determined through the UV-Vis spectra of the solutions (**Fig. R2b**). The calibration curve was established relating the concentration of H₂O₂ with the obtained absorbance, as shown in **Fig. R2c**. We collected the electrolyte after the OER test of Ru/TiO_x and conducted absorbance testing under the same conditions. We found that the absorbance values of Sample 1 and 2 at around 410 nm were 0.00001 and 0.00003 respectively (**Table R2**), and the concentration of H₂O₂ were calculated to be 0.011 mM according to the Equation (8). Therefore, we proved experimentally that H₂O₂ was not generated during the OER process.

$$Abs = 0.7664C_{H_2O_2} - 0.00886 \quad (8)$$

Fig. R2. a, Optical photo of the standard solution mixed with titanium sulfate and different concentrations of H₂O₂. **b**, The UV-Vis absorption spectra of standard solutions and electrolytes after OER test (Sample 1 and 2). **c**, Calibration curves derived from the UV-Vis absorbance at 410 nm.

Table R2. Concentration of H₂O₂ and absorbance (at 410 nm) of each solution.

Sample	C _{H₂O₂} (mM)	Absorbance (a.u.)
Standard 1	4.48	3.42461
Standard 2	2.80	2.11813
Standard 3	0.56	0.41919
Standard 4	0.11	0.09279
Standard 5	0.056	0.04289
Sample 1	0.011	0.00001
Sample 2	0.011	0.00003

References cited in this section are listed as bellow.

1. Hu, X. et al. Engineering nonprecious metal oxides electrocatalysts for two-electron water oxidation to H₂O₂. *Adv. Energy Mater.* **12**, 2201466 (2022).
2. Shi, X.-J. et al. Electrochemical synthesis of H₂O₂ by two-electron water oxidation reaction. *Chem* **7**, 38-63 (2021).
3. Baek, J. et al. Discovery of LaAlO₃ as an efficient catalyst for two-electron water electrolysis towards hydrogen peroxide. *Nat. Commun.* **13**, 7256 (2022).
4. Wang, Z.-L. et al. Single atomic Ru in TiO₂ boost efficient electrocatalytic water oxidation to hydrogen peroxide. *Sci. Bull.* **68**, 613-621 (2023).
5. Sandri, F. et al. Comparing catalysts of the direct synthesis of hydrogen peroxide in organic solvent: is the measure of the product an issue? *ChemCatChem* **13**, 2653-2663 (2021).
6. Kim, Y. et al. Revisiting the oxidizing capacity of the periodate-H₂O₂ mixture: identification of the primary oxidants and their formation mechanisms. *Environ. Sci. Technol.* **56**, 5763-5774 (2022).

Comment #3-4) The Tafel plots should be derived from the steady-state polarization curves (*ACS Energy Lett.* 2021, **6**, 1607).

Response: We deeply appreciate this important suggestion. According to the Tafel slope definition (Equation (9)), its value is supposed to be obtained with steady-state responses. By plotting the overpotential (η) against $\log j$, the slope ($2.303RT/\alpha nF$) and j_0 (from the intercept at equilibrium potential) are obtained.

$$\eta = (RT/\alpha nF)\ln j = (2.303RT/\alpha nF)\log j \quad (9)$$

According to the recommended reference (*ACS Energy Lett.* 2021, **6**, 1607), Tafel analysis should be performed with data acquired in a steady state and free of iR drop. Therefore, we re-analyzed the Tafel slopes of OER based on the as-reported method: 1) the 120th second of chronoamperometry responses were acquired at various overpotentials in the catalytic turnover region; 2) Tafel slopes were obtained by plotting the overpotential (η) against $\log j$. As a result, the derived Tafel slopes were 49.8, 82.8, 108.4 and 177.9 mV dec⁻¹ on the as-prepared Ru/TiO_x, annealed RuO_x/TiO₂, commercial RuO₂ and commercial IrO₂, respectively. The values are generally consistent with those obtained from the iR -free LSV curves, although there are small variations. The as-prepared Ru/TiO_x still showed better OER kinetics than the control samples. Accordingly, we have discussed the results (*Fig. 2b*, Page 11 and 13) and supplemented the test details in the Supplementary Information (*Supplementary Fig. 15*, Page 13) as below:

On page 10: “In order to evaluate the catalytic kinetics of OER, Tafel plots are obtained based on the *iR*-free LSV curves and the steady-state polarization curves (Fig. 2b and Supplementary Fig. 15)¹⁴, where Ru/TiO_x exhibits the lowest Tafel slope of 45.6 (49.8) mV dec⁻¹, indicating the highest charge transfer efficiency and fastest reaction rate among these prepared samples.”

References

14. Anantharaj, S., Noda, S., Driess, M. & Menezes, P. W. The pitfalls of using potentiodynamic polarization curves for tafel analysis in electrocatalytic water splitting. *ACS Energy Lett.* **6**, 1607-1611 (2021).

On page 34:

Fig. 2 | Electrochemical oxygen evolution reaction (OER) activity in 0.5 M H₂SO₄ solutions. *a*, OER polarization curves (both *iR*-corrected and *iR*-free), *b*, Tafel plots derived from the LSV curves (solid line) and the steady-state polarization curves (scatters) (values in parentheses were derived from steady-state polarization curves) and *c*, electrochemical impedance spectroscopy (EIS) of Ru/TiO_x, annealed RuO_x/TiO₂, commercial RuO₂/TiO₂, commercial IrO₂/TiO₂ and TiO₂. (Com. RuO₂/TiO₂ and com. IrO₂/TiO₂ denotes commercial RuO₂/TiO₂ and commercial IrO₂/TiO₂, respectively). *d*, Comparison of overpotentials *without iR correction* at 10 and 100 mA cm⁻² for Ru/TiO_x, annealed RuO_x/TiO₂, com. RuO₂/TiO₂ and com. IrO₂/TiO₂. *e*, Comparison of the overpotentials of Ru/TiO_x and state-of-the-art Ru/ Ir-based electrocatalysts at 10 mA cm⁻² in acidic media.

On SI page 13:

Supplementary Figure 15. Chronoamperometry responses of activity stabilized Ru/TiO_x (a), annealed RuO_x/TiO₂ (c), com. RuO₂ (e) and com. IrO₂ (g) in 0.5 M H₂SO₄. The corresponding steady-state polarization curves (Tafel plots) of Ru/TiO_x (b), annealed RuO_x/TiO₂ (d), com. RuO₂ (f) and com. IrO₂ (h) constructed from OER current densities sampled from steady-state chronoamperometry responses.

Comment #3-5) The applied potentials should be reported for the EIS measurements in Fig. 2c.

Response: We thank the reviewer for the constructive comment. Nyquist plots of electrochemical impedance spectroscopy (EIS) measurements were collected in frequency range of 100 kHz to 0.01 Hz at open-circuit potential with an amplitude of 5 mV AC voltage in 0.5 M H₂SO₄ solution. The details of the EIS measurements have been added to the revised manuscript (Page 24) as below:

On page 24: “Nyquist plots of electrochemical impedance spectroscopy (EIS) measurements were collected in frequency range of 100 kHz to 0.01 Hz at open-circuit potential with an amplitude of 5 mV AC voltage in 0.5 M H₂SO₄ solution.”

Comment #3-6) The OER durability of Ru/TiO_x needs to be further checked under the cycling condition (at least up to 200 mA cm⁻²). Could the authors measure more than 50 OER cycles up to 500 mA cm⁻²?

Response: Based on the reviewer's important suggestion, the OER stability of as-synthesized Ru/TiO_x was further evaluated by continuous OER cyclic voltammograms (CV) for 50 cycles up to 550 mA cm⁻². As shown in *Supplementary Fig. 18*, no obvious decay in polarization curves was observed for Ru/TiO_x after

50 OER cycles up to 550 mA cm^{-2} , suggesting excellent durability. The related CV test details and analysis of results have been added to the revised manuscript (Page 11) and Supplementary Information (*Supplementary Fig. 18*, Page 15) as below:

On page 11: “Thus, the OER stability of the as-prepared Ru/TiO_x was evaluated by continuous cyclic voltammograms (CVs) up to 550 mA cm^{-2} for 50 cycles and chronopotentiometry test at constant current densities of 10 mA cm^{-2} and 100 mA cm^{-2} . As shown in *Supplementary Fig. 18*, no obvious decay in polarization curves was observed for Ru/TiO_x after 50 OER cycles up to 550 mA cm^{-2} , suggesting excellent durability under large current densities.”

On page 24: “The OER stability was evaluated by continuous cyclic voltammograms (CVs) up to 550 mA cm^{-2} for 50 cycles and chronopotentiometry test at constant current densities of 10 mA cm^{-2} and 100 mA cm^{-2} .”

On SI page 15:

Supplementary Figure 18. CVs of the as-prepared Ru/TiO_x up to 550 mA cm^{-2} for 50 cycles (a) and polarization curves of Ru/TiO_x before and after 50 CV cycles.

Comment #3-7) Is the stability test under constant current density also performed in an undivided cell? Please check it. Because the dissolved Ru cations from the catalyst will be readily reduced and redeposited on the counter electrode in an undivided cell. Thus, if an undivided cell is used, the detected Ru ions content in electrolyte after the stability test shown in Fig. 3d (upper plot) cannot be trusted?

Response: We thank the reviewer very much for pointing this out. We acknowledge that Ru species tend to be readily reduced and redeposited on the counter electrode during the stability test. Therefore, the chronopotentiometric tests of the samples under a constant OER current density of 10 and 100 mA cm^{-2} were conducted in an H-type water electrolysis cell with the anode and cathode separated by a Nafion 117 membrane. ICP-MS measurements were carried out to detect the amounts of dissolved Ru in the anode side which ensure the reliability of the dissolved Ru content in the electrolyte. The relevant test details have been supplemented in the revised manuscript (Page 24) and Supplementary Information (*Supplementary Fig. 19*, Page 15) as below:

On page 24: “The chronopotentiometric tests of the samples under a constant OER current density of 10 and 100 mA cm^{-2} were conducted in an H-type water electrolysis cell with the anode and cathode separated

by a Nafion 117 membrane.”

On SI page 15:

Supplementary Figure 19. Electrocatalytic OER stability in 0.5 M H₂SO₄. Chronoamperometric curve obtained at a current density of 10 mA cm⁻² for the as-prepared Ru/TiO_x and the annealed RuO_x/TiO₂ in 0.5 M H₂SO₄. A photograph of a homemade H-type cell is shown in the inset, in which the anode and cathode sides are separated by a Nafion 117 membrane.

Comment #3-8) Considering that XPS probes the top few nanometers within a micron region of materials, the change of ruthenium content determined by XPS before and after the OER test is meaningless (Fig. 3d, lower plot)

Response: We appreciate the important comment of the reviewer very much. Since the electrocatalytic reaction mainly occurs at the catalyst surface (*Adv. Mater.* **35**, 2210565 (2023); *Joule* **5**, 1704-1731 (2021); *Adv. Mater.* **33**, 2004243 (2021)), we used XPS to reveal the changes in Ru content on the catalyst surface in the previous version of our manuscript. Considering that the ICP-MS test can measure the mass of Ru in the catalyst more accurately, we supplemented the ICP-MS measurements of catalysts after the OER stability test according to the reviewer's suggestions. As a result, the weight percentage of Ru in the as-prepared Ru/TiO_x catalyst slightly decreased from 0.075 wt% to 0.073 wt%, that is, 97.3% of Ru remained in the catalyst after OER stability test. For comparison, only 42.2% and 20.7% Ru remained in the annealed RuO_x/TiO₂ and commercial RuO₂/TiO₂ catalyst, proving the stability of the Ru sites in the Ru/TiO_x catalyst, consistent with the trend obtained by XPS results. Accordingly, the relevant analysis has been supplemented in the revised manuscript (*Fig. 3d*, Page 12 and 36) and Supplementary Information (Supplementary Fig. 20 and Table 2, Page) as below:

On page 12: “Inductively coupled plasma mass spectrometry (ICP-MS) analysis and X-ray photoelectron spectroscopy (XPS) were further performed to determine the amounts of dissolved Ru ions in the electrolytes after the stability test and the Ru content remained in the catalysts (*Fig. 3d*). For the as-synthesized Ru/TiO_x, the extremely low Ru ion concentration (6.4 ppb) in the electrolyte after the stability test and the maintenance of the Ru content (97.3%) in the catalyst manifest the effective protection of Ru sites from dissolution (*Fig. 3d* and Supplementary Table 2). For comparison, only 42.2% and 20.7% Ru remained in the annealed RuO_x/TiO₂ and commercial RuO₂/TiO₂ catalyst, proving the stability of the Ru sites in the Ru/TiO_x catalyst, consistent with the trend obtained by XPS results (Supplementary Fig. 20 and Table 2).”

On page 35:

Fig. 3 | Electrocatalytic OER stability in 0.5 M H₂SO₄ solutions. **a**, Chronoamperometric curves of Ru/TiO_x, annealed RuO_x/TiO₂, com. RuO₂/TiO₂ and com. IrO₂/TiO₂ for OER at 100 mA cm⁻². **b**, Chronoamperometric curves of Ru/TiO_x for OER at 10 mA cm⁻². **c**, Comparison of the overpotential and stability time of Ru/TiO_x with state-of-the-art OER electrocatalysts in acidic media. **d**, *Inductively coupled plasma-mass spectrometry (ICP-MS) analysis for dissolved Ru ions in post chronopotentiometry electrolyte and Ru mass percentage retained in Ru/TiO_x, annealed RuO_x/TiO₂ and com. RuO₂/TiO₂ catalyst after the chronoamperometric test.*

On SI page 16:

Supplementary Figure 20. Ru content of different electrocatalyst before and after OER stability test. Ru content in electrocatalyst before and after OER stability test determined by inductively coupled plasma-mass spectrometry (ICP-MS) (a) and XPS (b).

On SI page 33:

Supplementary Table 2. The mass loading (mg cm^{-2}) and weight percent ($\text{wt}\%$) of Ru in different samples (by ICP-MS measurement and EDS) and atomic percent ($\text{at}\%$) by XPS measurement.

Noble metal in sample	Mass loading (mg cm^{-2})	Weight % ($\text{wt}\%$) (ICP)	Weight % ($\text{wt}\%$) (EDS)	Atomic % ($\text{at}\%$) (XPS)
Ru/TiO _x	0.0715	0.075	7.7%	9.3%
Annealed RuO _x /TiO ₂	0.0867	0.083	8.2%	9.2%
Com. RuO ₂ /TiO ₂	0.0992	0.092	-	10.1%
Com. IrO ₂ /Nb ₂ O ₅	0.1135	0.107	8.8%	7.5%
Ru/TiO_x after OER	0.0695	0.073	6.4%	7.2%
Annealed RuO_x/TiO₂ after OER	0.0367	0.035	3.3%	4.3%
Com. RuO₂/TiO₂ after OER	0.0201	0.019	-	2.4%

*Note: Since the catalysts are binder-free electrodes, the catalysts are dissolved together with the substrates during the ICP test, while EDS and XPS only detect the surface content, so the $\text{wt}\%$ obtained through ICP is less than that obtained through EDS and XPS, but the trend is consistent.

Comment #3-9) The deconvolution of XPS spectra in Figs. 2a-c is very slipshod. There lacks consistent full width at half maximum and spin-orbit splitting of Ru 3d. The C 1s spectrum at 284.8 eV overlapped with Ru 3d_{3/2} is also not reported. Please use a supplementary table to summarize the fit parameters of XPS spectra.

Response: We thank the reviewer for the constructive comment. We refitted all XPS data according to the peak fitting rules (*Nat. Catal.* **4**, 1012-1023 (2021); *J. Vac. Sci. Technol. A* **40**, 063201 (2022); <http://srdata.nist.gov/xps/selectEnergyType.aspx>) and summarized the **fitting parameters (including peak position, full-width at the half of the maximum and peak area)** into the supplementary tables as suggested. In addition, we also supplemented *in-situ* XPS testing to reveal the chemical environments of Ti and Ru under actual reaction conditions. The obtained results were also fitted in the same way and the parameters were reported. Particularly, we pay special attention to the Ru 3d peak splitting issue raised by the reviewer. We have marked the C 1s peak in each Ru 3d spectrum and reported its parameters in the supplementary tables. Accordingly, the XPS analysis has been revised (*Fig. 4a-c*, Page 12 and 37) and the fitting parameters have been supplemented in the Supplementary Information (*Supplementary Figs. 26-29* and *Table 8-9*).

On page 12: “To further verify the high stability of Ru/TiO_x for OER in acidic electrolyte, the chemical states for Ru, Ti and O in the three samples after the acidic OER stability tests were analyzed and compared with those before OER (*Fig. 4c*, *Supplementary Table 8*). For Ru/TiO_x, the peak at 280.61 eV for Ru 3d_{5/2} (Ru⁰) remained generally unchanged as compared with that of the sample before the OER test (280.60 eV), while the peak at 281.2 eV for Ru 3d_{3/2} (Ruⁿ⁺, n<4) slightly shifted to higher binding energies, indicating that the active Ru sites were only partially oxidized but mainly remained in the low-valence state (Ruⁿ⁺, n<4) during the 50 h OER test (*Fig. 4c*). For comparison, the changes for annealed RuO_x/TiO₂ and commercial RuO₂/TiO₂ are much more significant: the peaks for Ru⁰ and Ru⁴⁺ shift positively for 0.54 and 0.90 eV, respectively, indicating that the active species Ru were all over-oxidized to Ruⁿ⁺ (n>4), which were easily separated from the catalysts and dissolved during the reaction process, leading to the degradation

of the OER performance. The Ru^{n+}/Ru^0 ratios calculated from the corresponding peak area in the Ru 3d XPS spectra indicate that the oxidation state of Ru in the as-prepared catalysts follows the trend of commercial $RuO_2/TiO_2 > \text{annealed } RuO_x/TiO_2 > Ru/TiO_x$, which is opposite to the trend of OER stability (Supplementary Fig. 26). The established valence-stability relationship proved that the low-valence Ru in Ru/TiO_x , due to the strong interaction between Ru and TiO_x , is highly active and stable, further highlighting the advantages of the in situ and one-step growth strategy. Afterwards, the chemical environments of Ti and Ru in the Ru/TiO_x catalyst under actual OER conditions were monitored by in-situ XPS (Supplementary Figs. 27-29 and Table 9) Significantly, the Ru 3d XPS peaks at 280.1 and 284.2 eV exhibit negligible changes with the applied potential increased from 1.0 to 1.7 V vs. RHE (Supplementary Fig. 28a and Table 9). Detailed quantitative analysis shows the ratio of Ru^{n+}/Ru^0 remains almost identical at 0.7 as the voltage increases, which is consistent with the ex-situ XPS analysis results (Supplementary Figs. 26 and 29). The above results further confirm the stable Ru chemical state in Ru/TiO_x during the OER process. For the Ti 2p spectra (Supplementary Figs. 28b and Table 9), the coexistence of Ti^{3+} and Ti^{4+} species was distinguished. Interestingly, as the voltage increases, the ratio of Ti^{3+}/Ti^{4+} slightly decreases and remains stable at 0.17, indicating the critical role of the Ti^{3+} in stabilizing Ru active sites. Corresponding to this phenomenon is the O_V/O_L (oxygen vacancy/lattice oxygen) ratio obtained from the O 1s spectra (Supplementary Figs. 28c and 29). It can be seen that as the voltage increases, the O_V/O_L ratio first decreases and then almost returns to the initial state, manifesting that O_V regeneration is accompanied by the release of oxygen, thereby stabilizing active species³⁹.”

On page 36:

Fig. 4 | Electronic structure analysis of Ru/TiO_x . *a*, XPS of Ti 2p and Ru 3p for the bare TiO_2 and Ru/TiO_x . *b,c*, Ru 3d XPS spectra of Ru/TiO_x , annealed RuO_x/TiO_2 and com. RuO_2/TiO_2 before OER stability test (*b*) and after OER stability test (*c*). *d*, Ru K-edge synchrotron-based X-ray absorption near-edge structure

(XANES) spectra of Ru/TiO_x before and after OER stability test using Ru foil and commercial RuO₂ as references. **e**, Fourier-transformed (FT) k^3 -weighted $\chi(k)$ -function of the extended X-ray absorption fine structure (EXAFS) spectra for the Ru K-edge. **f**, Relation between the Ru K-edge absorption energy (E_0) and valence states for Ru/TiO_x, Ru/TiO_x after OER stability test, Ru foil and RuO₂. **g-j**, Wavelet transforms for the k^3 -weighted EXAFS signals of Ru foil (**g**), RuO₂ (**h**), Ru/TiO_x (**i**) and Ru/TiO_x after OER stability test (**j**).

On SI page 20:

Supplementary Figure 28. In-situ XPS spectra of Ru/TiO_x during the OER. In-situ Ru 3d (a), Ti 2p & Ru 3p (b) and O 1s (c) XPS spectra recorded of the as-prepared Ru/TiO_x at applied potential during 1.0-1.7 V vs. RHE.

Supplementary Figure 29. Variation of Ru^{III+}/Ru⁰, Ti³⁺/Ti⁴⁺ and O_v/O_L (oxygen vacancy/lattice oxygen) ratio from in-situ XPS measurement.

On SI page 38:

Supplementary Table 8. High resolution Ru 3d XPS peak fitting parameters of different samples before and after OER

Sample	Core level	Peak position (eV)	Peak area	FWHM (eV) ^{a)}
Ru/TiO _x	Ru 3d _{5/2}	280.6	34594.87	0.68
		281.16	25185.03	1.57
		284.7	23063.65	0.93
	Ru 3d _{3/2}	285.25	16790.02	1.81
	C 1s	284.8	27369.31	1.81
Annealed RuO _x /TiO ₂	Ru 3d _{5/2}	280.62	67103.88	0.98
		282.33	55862.16	1.78
		284.72	44735.92	0.98
	Ru 3d _{3/2}	286.43	37241.44	1.82
	C 1s	284.8	34397.01	1.77
Com. RuO ₂ /TiO ₂	Ru 3d _{5/2}	280.65	14552.20	0.92
		282.43	15022.85	1.82
		284.75	9701.47	1.02
	Ru 3d _{3/2}	286.53	10015.23	1.85
	C 1s	284.8	6377.99	1.91
Ru/TiO _x after OER	Ru 3d _{5/2}	280.61	21619.56	1.41
		282.33	15987.43	1.91
		284.72	14413.04	1.58
	Ru 3d _{3/2}	286.44	10658.29	1.9
	C 1s	284.8	9130.65	1.98
Annealed RuO _x /TiO ₂ after OER	Ru 3d _{5/2}	281.16	14047.01	0.95
		282.38	15072.05	1.81
		285.26	9364.67	1.05
	Ru 3d _{3/2}	286.48	10048.03	1.89
	C 1s	284.8	28387.34	1.5
Com. RuO ₂ /TiO ₂ after OER	Ru 3d _{5/2}	281.55	12506.05	1.23
		282.71	16144.38	1.47
		285.65	8337.36	1.26
	Ru 3d _{3/2}	286.8	10762.92	1.65
	C 1s	284.8	56728.58	1.27

^{a)} FWHM: full-width at the half of the maximum.

On SI page 39:

Supplementary Table 9. High resolution Ru 3d, Ti 2p and O 1s XPS peak fitting parameters of Ru/TiO_x at applied potential during 1.0-1.7 V vs. RHE.

Sample	Core level	Peak position (eV)	Peak area	FWHM (eV) ^{a)}
Ru/TiO _x -1.0	Ru 3d _{5/2}	280.12	2593.16	1.07
		280.77	1836.78	1.17
	Ru 3d _{3/2}	284.22	1728.77	1.11
		284.87	1224.52	1.19
	Ti 2p _{3/2}	457.90	7399.48	1.20
		458.51	23726.30	1.14
	Ti 2p _{1/2}	463.90	3699.74	1.21
		464.51	11863.15	1.54
	Ru 3p _{3/2}	460.60	1470.22	1.90
		462.96	2533.76	1.48
	O 1s	529.52	19079.02	1.88
		530.45	9727.79	1.45
		531.47	8456.91	1.79
Ru/TiO _x -1.2	Ru 3d _{5/2}	280.13	2734.80	1.05
		280.81	1969.02	1.20
	Ru 3d _{3/2}	284.23	1823.20	1.07
		284.91	1312.68	1.21
	Ti 2p _{3/2}	457.90	7001.99	1.23
		458.58	22629.02	1.09
	Ti 2p _{1/2}	463.90	3500.99	1.23
		464.50	11314.51	1.49
	Ru 3p _{3/2}	460.60	1591.89	1.89
		462.95	2534.37	1.68
	O 1s	529.55	19282.80	1.86
		530.40	9066.99	1.51
		531.43	8160.46	1.89
Ru/TiO _x -1.4	Ru 3d _{5/2}	280.15	2678.28	1.07
		280.83	1945.85	1.44
	Ru 3d _{3/2}	284.25	1785.52	1.08
		284.93	1297.23	1.45
	Ti 2p _{3/2}	457.90	3294.47	1.16
		458.54	15788.83	1.14
	Ti 2p _{1/2}	463.90	1647.24	1.16
		464.54	7894.42	1.54
	Ru 3p _{3/2}	460.90	1172.22	1.98
		463.08	1976.27	1.67
	O 1s	529.60	20864.25	1.72
		530.48	9731.28	1.41
		531.49	8407.48	1.98
Ru/TiO _x -1.6	Ru 3d _{5/2}	280.17	2435.83	1.03
		280.95	1688.01	1.37

Ru/TiO _x -1.7	Ru 3d _{3/2}	284.27	1557.22	1.05
		285.05	1218.68	1.37
	Ti 2p _{3/2}	457.90	5249.07	0.98
		458.60	29718.37	1.10
	Ti 2p _{1/2}	463.90	2624.53	0.98
		464.60	14859.18	1.50
	Ru 3p _{3/2}	460.40	2316.29	1.92
		463.15	3039.33	1.52
	O 1s	529.67	21524.80	1.63
		530.48	9988.22	1.38
	Ru 3d _{5/2}	531.52	7457.96	1.89
		280.21	2506.04	1.04
	Ru 3d _{3/2}	281.00	1826.04	1.35
		284.31	1670.69	1.04
	Ti 2p _{3/2}	285.10	1217.36	1.35
		457.90	5156.78	1.12
	Ti 2p _{1/2}	458.63	29558.12	1.07
		463.90	2578.39	1.12
	Ru 3p _{3/2}	464.63	14779.06	1.47
		460.60	2411.36	1.98
O 1s	463.27	3225.89	1.89	
	526.69	23711.47	1.56	
	530.46	11822.26	1.27	
	531.50	8267.88	2.03	

^{a)}FWHM: full-width at the half of the maximum.

Comment #3-10) In general, higher oxidation state of Ru-sites leads to better activity (*J. Phys. Chem. C* 2017, **121**, 18516–18524; *ACS Catal.* 2020, **10**, 12182-12196). The lower the oxidation state of Ru will tend to result in stronger OH* binding and higher overpotentials. This is in stark contrast to what the authors find. The possible OER mechanism needs to be discussed among many mechanisms suggested in recent literatures.

Response: We thank the reviewer for the insightful comments. Based on the literatures recommended by the reviewer (*J. Phys. Chem. C* 2017, **121**, 18516-18524; *ACS Catal.* 2020, **10**, 12182-12196) and other literatures on theoretical calculations studying the OER process (*ChemCatChem* 2011, **3**, 1159-1165; *Nat. Catal.* 2020, **3**, 516-525; *Chem Catal.* 2021, **1**, 258-271), the energies of adsorbed OER intermediates (*OH, *O, and *OOH) scale with one another. Therefore, it is not comprehensive enough to analyze OER activity using only the OH* binding. Actually, $\Delta G_{O} - \Delta G_{OH}$ has been widely acknowledged as a descriptor for OER activity. It has been shown previously that this descriptor correlates reasonably well with experimental activity (*Science* 2017, **355**, eaad4998; *J. Phys. Chem. C* 2017, **121**, 18516-18524; *ACS Catal.* 2020, **10**, 12182-12196; *ChemCatChem* 2011, **3**, 1159-1165). Herein, we apply $\Delta G_{O} - \Delta G_{OH}$ as additional justification for structure-activity relationship. To draw the correlations between the oxidation state of Ru metal atoms and the OER activity, we constructed models with different oxidation states of Ru (namely, P-RuO_{1.6}/TiO₂, P-RuO/TiO₂ and V-RuO/TiO_x, here P denotes the perfect structure and V denotes vacancy structure). Then, the $\Delta G_{O} - \Delta G_{OH}$ descriptor as a function of Ru bader charge partitioning scheme was established (Fig. R2).

It has been reported that larger $\Delta G_{\text{O}}-\Delta G_{\text{OH}}$ descriptor values translate to higher catalytic activity (*J. Phys. Chem. C* 2017, **121**, 18516-18524; *ACS Catal.* 2020, **10**, 12182-12196). As shown in **Fig. R3**, the $\Delta G_{\text{O}}-\Delta G_{\text{OH}}$ value increases with the increase of Ru Bader charge, indicating that Ru in the V-RuO/TiO_x sample possesses the highest OER activity, consistent with the η_{calc} results. This activity enhancement originates from the strong electronic interaction between Ru and V-TiO_x, which optimizes the adsorption energy of OER intermediates (*OH, *O, and *OOH) and reduces the energy barrier of the rate-determine step.

Fig. R3. *a*, $\Delta G_{\text{O}}-\Delta G_{\text{OH}}$ binding energy OER activity descriptor versus Ru Bader charge. *b*, Theoretical overpotential (η_{calc}) values versus Ru Bader charge.

Secondly, the possible OER mechanism has been re-analyzed. Since the adsorbate evolution mechanism (AEM) pathway involves four electron-proton transfer steps, it is characterized by pH-independent activity on the reversible hydrogen electrode (RHE) scale (*Adv. Mater.* 10.1002/adma.202305939; *Adv. Mater.* 2023, **35**, 2210565; *ACS Catal.* 2023, **13**, 256-266; *Energy Environ. Sci.* 2022, **15**, 2356). In contrast, the lattice oxygen-evolution mechanism (LOM) pathway involves non-concerted proton-electron transfers and therefore exhibits pH-dependent activity (*Nat. Chem.* 2017, **9**, 457-465; *Nat. Commun.* 2022, **13**, 4871). To further explore the possible catalytic mechanism on Ru/TiO_x, the pH-dependence measurements of the corresponding OER activities were performed in the pH range of 0.3-1. As a result, the Ru/TiO_x shows pH-independent OER kinetics on the RHE scale, which is typical for AEM pathway and consistent with the mechanism analysis in our original manuscript. The following description has been added to the revised manuscript (Page 18 and 25) and Supplementary Information (*Supplementary Fig. 35*, Page 25) as below:

On page 18: “To further explore the possible catalytic mechanism on Ru/TiO_x, the pH-dependence measurements of the corresponding OER activities were performed. As a result, the Ru/TiO_x shows pH-independent OER kinetics on the RHE scale, typical for AEM pathway (*Supplementary Fig. 35*).”

On page 25: “Typically, the pH-dependence measurement was carried out at 1.23 to 1.53 V vs. RHE in H₂SO₄ with different pH (0.3, 0.4, 0.7 and 1).”

Supplementary Figure 35. a, OER activity of Ru/TiO_x with varying pH. **b**, pH dependence on the OER potential at different current densities for Ru/TiO_x.

Comment #3-11) In Figs. 5d-e, the DFT results are reported on the base of two O-adsorbed structures, V-RuO/TiO_x and P-RuO_{1.6}/TiO₂, rather than the pristine Ru/TiO₂ and Ru/TiO_x. It is unclear why the authors treated it this way. The reasons should be provided.

Response: We appreciate the important comment of the reviewer very much. In the ‘Mechanism analysis of OER activity’ part, we firstly discussed the Ru-Ru bond length and ΔE_O in P-Ru/TiO₂ and V-Ru/TiO_x models (Figs. 5a-b). The main conclusion drawn is that the existence of vacancies improves the stability and antioxidant capacity of the V-Ru/TiO_x structure. In the following analysis of the OER mechanisms (Figs. 5c-e), the optimized V-RuO/TiO_x and P-RuO_x/TiO₂ structure were applied. The main reason is that during the OER process, the amorphous nonstoichiometric oxide layers (RuO_x) will be generated at the catalyst/electrolyte interface through an electro-oxidation process (*Science* 2016, **353**, 1011-1014; *Nat. Commun.* 2019, **10**, 4849; *J. Phys. Chem. C* 2019, **123**, 22151-22157). Since the generated oxides are nonstoichiometric, we constructed models (V-RuO/TiO_x, P-RuO/TiO₂ and P-RuO_x/TiO₂) with different oxidation states of Ru to analyze their adsorption behaviors for intermediates during the OER process. According to the reviewers' suggestions, we have added the corresponding description to the revised manuscript (Page 20) as below:

On page 20: “It has been reported that OER catalysts tend to pre-adsorb oxygen species and generate amorphous nonstoichiometric oxide layers under the OER working condition^{13,41}. Therefore, the Ru₅ cluster will also be present as oxides during OER. Since the existence of oxygen vacancies will affect the oxidation state of Ru, we constructed different models (V_{1O}-RuO/TiO_x, V_{2O}-RuO/TiO_x, P-RuO/TiO₂ and P-RuO_x/TiO₂, V_{2O} denotes two V_O) to study the OER process (Supplementary Fig. 40).”

REVIEWERS' COMMENTS

Reviewer #1 (Remarks to the Author):

All my concerns are well addressed, and the manuscript is suggested to be accepted.

Reviewer #2 (Remarks to the Author):

The authors have addressed all my concerns. I now support the publication of this manuscript in Nature Communications.

Reviewer #3 (Remarks to the Author):

The authors have responded satisfactorily to the comments. The article presents an interesting material, and the work is well described. I recommend publication as is.